# Building resilient cervical cancer prevention through gender-neutral HPV vaccination

Irene Man[1], Damien Georges[1], Rengaswamy Sankaranarayanan[2], Partha Basu[1], Iacopo Baussano[1]*

[1]International Agency for Research on Cancer (IARC/WHO), Early Detection, Prevention and Infections Branch, Lyon, France; [2]Karkinos Healthcare, Ernakulam, India

**Abstract** The COVID-19 pandemic has disrupted HPV vaccination programmes worldwide. Using an agent-based model, EpiMetHeos, recently calibrated to Indian data, we illustrate how shifting from a girls-only (GO) to a gender-neutral (GN) vaccination strategy could improve the resilience of cervical cancer prevention against disruption of HPV vaccination. In the base case of 5-year disruption with no coverage, shifting from GO to GN strategy under 60% coverage (before disruption) would increase the resilience, in terms of cervical cancer cases still prevented in the disrupted birth cohorts per 100,000 girls born, by 2.8-fold from 107 to 302 cases, and by 2.2-fold from 209 to 464 cases under 90% coverage. Furthermore, shifting to GN vaccination helped in reaching the World Health Organization (WHO) elimination threshold. Under GO vaccination with 60% coverage, the age-standardised incidence rate of cervical cancer in India in the long term with vaccination decreased from 11.0 to 4.7 cases per 100,000 woman-years (above threshold), as compared to 2.8 cases (below threshold) under GN with 60% coverage and 2.4 cases (below threshold) under GN with 90% coverage. In conclusion, GN HPV vaccination is an effective strategy to improve the resilience to disruption of cancer prevention programmes and to enhance the progress towards cervical cancer elimination.

*For correspondence: baussanoi@iarc.who.int

## Editor's evaluation

This study presents valuable findings on how gender-neutral vaccination against human papillomavirus can help improve program resilience in the case of vaccination disruptions. The evidence supporting the claims of the authors is convincing, although the results are only applicable to India and other countries with a similar HPV context; researchers can adapt the model for their local context and use it as a starting point for future research.

## Introduction

In August 2020, the World Health Assembly adopted the Global Strategy for cervical cancer elimination, with the overarching target of reducing the age-standardised incidence rate (ASIR) of cervical cancer to fewer than 4 cases per 100,000 woman-years (*WHO, 2020a*). To reach this target, the following actions have been recommended: to fully vaccinate 90% of all girls with the HPV vaccine by age 15 years, to screen 70% of women using a high-performance test by the age of 35, and again by the age of 45, and to treat 90% of women with pre-cancer and 90% of women with invasive cancer (*WHO, 2020b*). However, the delivery of these interventions has been severely disrupted worldwide by the outbreak of the COVID-19 pandemic, which was declared a global emergency in the same year

by the World Health Organization (WHO) (*WHO, 2020a*). More specifically, HPV vaccination world-wide was affected at many different levels: (a) where population-based HPV vaccination programmes were already active, vaccine delivery slowed down or was interrupted (*Muhoza et al., 2021*), (b) the launch of HPV vaccination programmes was delayed in several countries, in particular in resource-limited settings (*The Lancet Oncology, 2022*), and (c) the production of HPV vaccines was also limited in favour of manufacturing other vaccines (*WHO, 2022a*).

Apart from the COVID-19 pandemic, which has resulted in extensive disruption of health-care provision at a global level, health-care systems or public-health programmes at a local level have also regularly suffered severe disruption caused by other factors such as changes in political commitment, financial constraints, scepticism of the civil society, geo-political unrest, and environmental disasters (*Colón-López et al., 2021*; *Gallagher et al., 2017*; *Germani et al., 2022*; *Jawad et al., 2021*; *Larson, 2020*; *McGrath, 2022*). Such circumstances may even lead key public health actors to reconsider their previous commitments. These partial or complete disruptions to organised disease prevention and control programmes may multiply public health crises (*Sharpless, 2020*), as well as provoke substantial, usually unquantified waste of human, logistic, and financial resources (*Richards et al., 2020*). Clearly, devising and implementing pre-emptive measures aimed at improving the resilience of public health programmes would help mitigate disruption.

In a previous study, conducted before the COVID-19 pandemic in a high-income country, we showed that the addition of vaccination in boys to a programme targeting girls only could increase the resilience of the vaccination programme (*Elfström et al., 2016*). In the present paper, we complement our previous work by assessing the potential effect of switching from girls-only (GO) to gender-neutral (GN) HPV vaccination on resilience and progress towards the elimination of cervical cancer in India, which in turn may help local health authorities in deciding on their vaccination strategy. Currently, India is the country with the largest burden of cervical cancer (*Bonjour et al., 2021*). Cervical cancer screening is still limitedly accessible in India, with less than 3% ever-in-life coverage in women aged 30–49 years (*Sankaranarayanan et al., 2019*; *The Lancet Oncology, 2022*). HPV vaccination (in girls) has been introduced in two Indian states with high vaccination coverage (*Sankaranarayanan et al., 2019*). Following the recent marketing authorisation granted to an indigenous vaccine (*The Lancet Oncology, 2022*), the prospect of introducing HPV vaccination into the national immunisation programme has also significantly improved.

## Results

To estimate the impact of HPV vaccination on cervical cancer burden, we used the agent-based HPV transmission model, EpiMetHeos (*Man et al., 2022*), in combination with the cervical cancer progression model, Atlas (*Bonjour et al., 2021*). Routine HPV vaccination at age 10 under the GO or GN strategy was modelled with different vaccination coverages in boys and girls from the range between

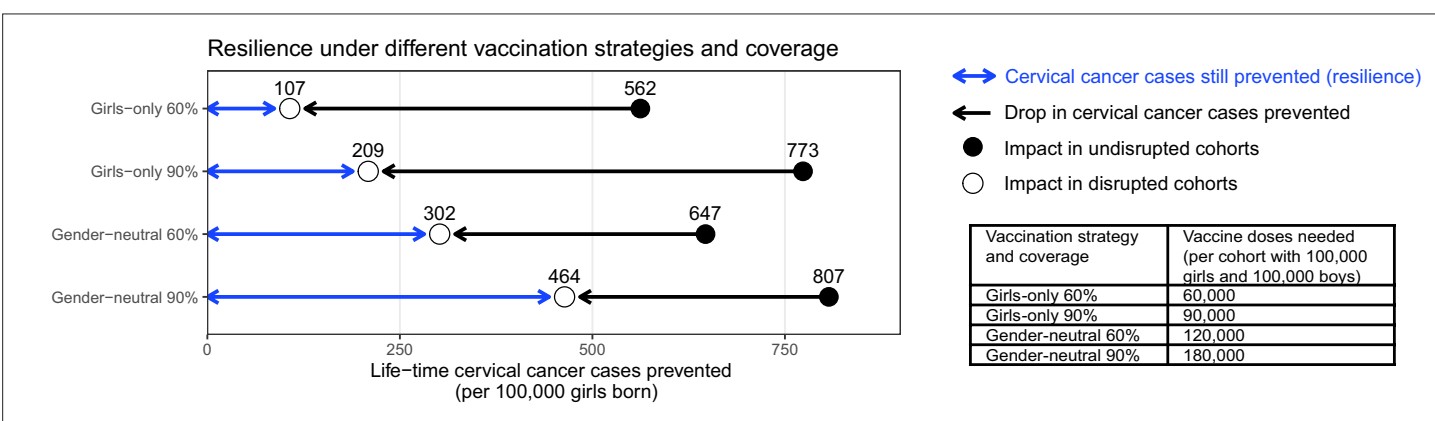

**Figure 1.** Resilience against HPV vaccination disruption in the base case. Predicted resilience, defined as life-time number of cervical cancer cases still prevented in the birth cohorts with disruption of vaccination per 100,000 girls born (blue arrow), and drop in cervical cancers prevented as compared to impact in the cohorts vaccinated prior to the disruption (black arrow), under the four highlighted scenarios. Disruption was simulated according to the base case with a period of disruption of 5 years and 0% coverage in girls and boys during the disruption period.

0% and 100%, at 10% intervals, as we do not yet know what coverage a national HPV immunisation programme in India may reach. Estimates of the life-time number of cervical cancer cases prevented per 100,000 girls born were derived by cohort.

### GN vaccination to improve resilience of cervical cancer prevention

To consider the impact of scaling up vaccination from such suboptimal coverage in girls, we highlighted the following four scenarios: (A) GO strategy with 60% coverage in girls, (B) GO strategy with 90% coverage in girls, (C) GN strategy with 60% coverage in both girls and boys, (D) GN strategy with 90% coverage in both girls and boys. As the base case, we simulated disruption in vaccination for 5 years, taking place 10 years after the introduction of vaccination. Life-time cervical cancer cases prevented in the birth cohorts vaccinated prior to the disruption period ranged between 562 (UI: 444, 676) and 807 (UI: 752, 853) cases per 100,000 girls born in the four highlighted scenarios. The highest impact was achieved by the GN strategy with 90% coverage, closely followed by GO with 90% coverage, then GN with 60% coverage, and then GO with 60% coverage (*Figure 1*).

Among the four highlighted scenarios, the GO strategy with 60% coverage also had the lowest resilience, which we defined as the mean number of cases still prevented across the birth cohorts with disruption (*Figure 1*). We found that the life-time number of cervical cancer cases prevented would drop considerably from 562 (UI: 444, 676) cases in the undisrupted cohorts to 107 (UI: 7, 214) cases per 100,000 girls born in the disrupted cohorts (i.e., resilience of 107). Increasing the coverage under the GO strategy from 60% to 90% would lead to a small absolute gain in resilience, from 107 (UI: 7, 214) to 209 (UI: 81, 340) cases prevented per 100,000 girls born, and this despite the larger increase in impact in the undisrupted cohorts vaccinated prior to the disruption from 562 (UI: 444, 676) to 773 (UI: 701, 836) cases prevented. By contrast, switching from GO to GN vaccination, under 60% coverage, led to a moderate gain of impact in the undisrupted cohorts from 562 (UI: 444, 676) to 647 (UI: 539, 746) cases prevented, but a substantial increase in resilience from 107 (UI: 7, 214) to 302 (UI: 170, 437) cases prevented per 100,000 girls born. Finally, GN vaccination with 90% coverage led to the highest impact

**Table 1.** Sensitivity analyses on coverage at disruption and duration of disruption on resilience.

Life-time number of cervical cancer cases prevented per 100,000 girls born in birth cohorts vaccinated prior to disruption in part I. Sensitivity analyses on coverage at disruption in part II and on duration of disruption in part III on resilience (defined as the life-time number of cervical cancer cases still prevented in the birth cohorts with disruption of vaccination per 100,000 girls born) and resilience ratio (defined as fold change in resilience by switching from one scenario to another). Uncertainty intervals are reported in brackets. [Resilience ratio of scenario X to scenario Y] is defined as [resilience of scenario Y]/[resilience of scenario X]. For example, in the base case, [resilience ratio of GO 60% to GO 90%] = [resilience of GO 90%]/[resilience of GO 60%]=209/107=2.0.

*I. Life-time number of cervical cancer cases prevented prior to disruption*

| Scenario | GO 60% | GO 90% | GN 60% | GN 90% |
|---|---|---|---|---|
| No disruption | 562 (444, 676) | 773 (701, 836) | 647 (539, 746) | 807 (752, 853) |

*II. Sensitivity analyses on coverage at disruption (with duration of disruption fixed at 5 years)*

| | Resilience by vaccination strategy and coverage | | | | Resilience ratio | | |
|---|---|---|---|---|---|---|---|
| Coverage at disruption in % | GO 60% | GO 90% | GN 60% | GN 90% | GO 60% to GO 90% | GO 60% to GN 60% | GO 90% to GN 90% |
| 0 (base case) | 107 (7, 214) | 209 (81, 340) | 302 (170, 437) | 464 (328, 602) | 2.0 | 2.8 | 2.2 |
| 20 | 271 (155, 391) | 355 (221, 490) | 425 (297, 559) | 550 (416, 680) | 1.3 | 1.6 | 1.6 |
| 40 | 410 (277, 534) | 476 (343, 599) | 527 (401, 647) | 621 (500, 730) | 1.2 | 1.3 | 1.3 |

*III. Sensitivity analyses on duration of disruption (with coverage at disruption fixed at 0%)*

| | Resilience by vaccination strategy and coverage | | | | Resilience ratio | | |
|---|---|---|---|---|---|---|---|
| Duration of disruption in years | GO 60% | GO 90% | GN 60% | GN 90% | GO 60% to GO 90% | GO 60% to GN 60% | GO 90% to GN 90% |
| 1 | 137 (26, 253) | 261 (125, 407) | 365 (215, 502) | 517 (372, 655) | 1.9 | 2.7 | 2.0 |
| 2 | 125 (17, 233) | 240 (105, 375) | 344 (206, 480) | 500 (359, 642) | 1.9 | 2.7 | 2.1 |
| 5 (base case) | 107 (7, 214) | 209 (81, 340) | 302 (170, 437) | 464 (328, 602) | 2.0 | 2.8 | 2.2 |
| 10 | 80 (0, 182) | 154 (33, 275) | 226 (96, 358) | 382 (240, 525) | 1.9 | 2.8 | 2.5 |

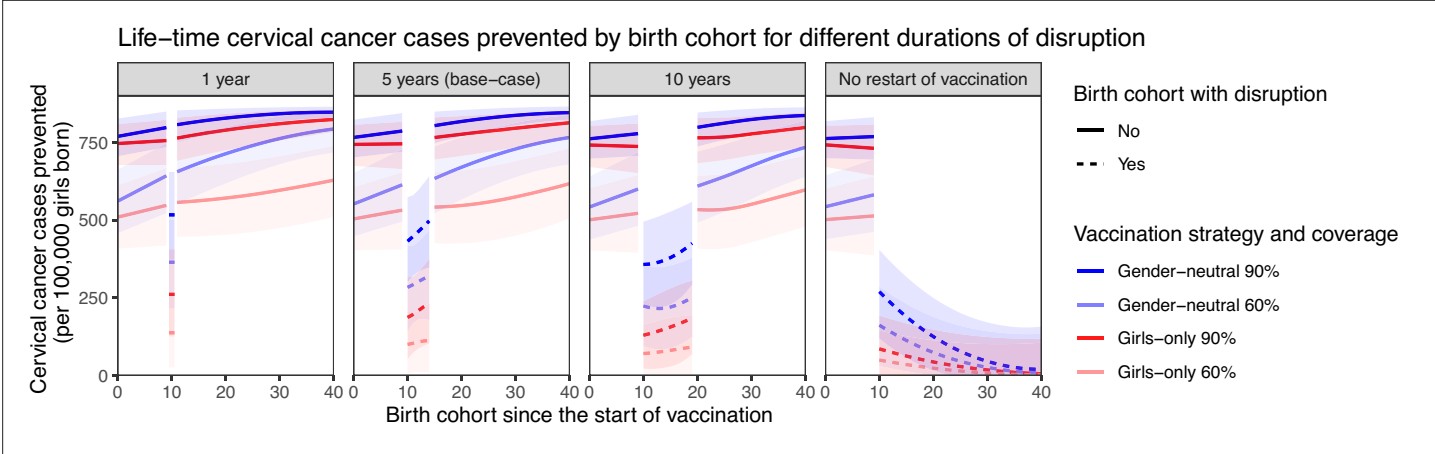

**Figure 2.** Resilience against HPV vaccination disruption in sensitivity analyses on duration of disruption. Predicted life-time cervical cancer cases prevented by birth cohorts (per 100,000 girls born) under different durations of disruption (panel) and in the four highlighted scenarios: girls-only strategy with 60% coverage (light red), girls-only strategy with 90% coverage (dark red), gender-neutral strategy with 60% coverage (light blue), and gender-neutral strategy with 90% coverage (dark blue). Vaccination coverage was fixed at 0% in girls and boys during the period of disruption. Birth cohort 0 corresponds to the first vaccinated cohort. Birth cohorts with disruption correspond to dashed lines and birth cohorts without disruption to solid lines.

of 807 (UI: 752, 853) cases prevented per 100,000 girls born in the undisrupted cohorts as well as the highest resilience of 464 (UI: 328, 602) cases prevented per 100,000 girls born in the disrupted cohorts.

Considering the required number of vaccine doses, we found a trade-off between dose-efficiency and resilience. In general, GO vaccination favoured dose-efficiency and GN vaccination favoured resilience (*Figure 1*). For instance, increasing GO coverage from 60% to 90% would lead to a higher impact in undisrupted cohorts, while needing less additional vaccine doses than switching from GO to GN vaccination under 60% coverage. However, this switch from GO to GN vaccination would yield a higher gain in resilience than when increasing GO coverage from 60% to 90% (2.8-fold versus 2-fold increase). Similarly, switching from GO to GN vaccination under 90% coverage would require an additional doubling of the number of doses and only marginally improve the impact in the undisrupted cohorts, but would produce a 2.2-fold gain in resilience.

The same trade-off between dose-efficiency and resilience was found when stratifying by Indian state. Interestingly, in the states with high cervical cancer incidence, we found slightly lower resilience under the GO strategy and a slightly greater relative gain in resilience when switching from GO to GN vaccination than in states with low incidence (*Supplementary file 3*, *Appendix 2—figures 1–3*). In the base case, for example, the resilience ratio for switching from GO to GN strategy (under 60% coverage) was 3-fold in states with high cervical cancer incidence and 2.7-fold in states with low cervical cancer incidence.

Sensitivity analyses on less strong disruption in vaccination coverage or alternative durations of disruption also yielded higher resilience with GN than GO vaccination (*Table 1*, *Appendix 2—figures 2–3*). The gain in resilience by switching from GO to GN vaccination was most evident under complete disruption of vaccination (i.e., 0% coverage at disruption) but still consistently found under less strong disruption (i.e., 20% or 40% coverage at disruption) (*Table 1* part II, *Appendix 2—figure 2*). As expected, resilience decreased with longer duration of disruption, with any vaccination strategy, but was always higher with the GN strategy (*Figure 2*). The gain in resilience by switching from GO to GN vaccination was consistently above 2.7-fold under 60% coverage and above 2.0-fold under 90% coverage (*Table 1* part III). Finally, in the scenario with no restart of vaccination, GN vaccination could ensure some resilience of at least 125 cases prevented per 100,000 girls born for eight birth cohorts from the start of disruption (*Figure 2*).

## GN vaccination to enhance progress towards elimination of cervical cancer

Subsequently, we assessed the feasibility of reaching the WHO elimination threshold for ASIR of cervical cancer under highlighted scenarios. Introducing GO vaccination with 60% coverage would

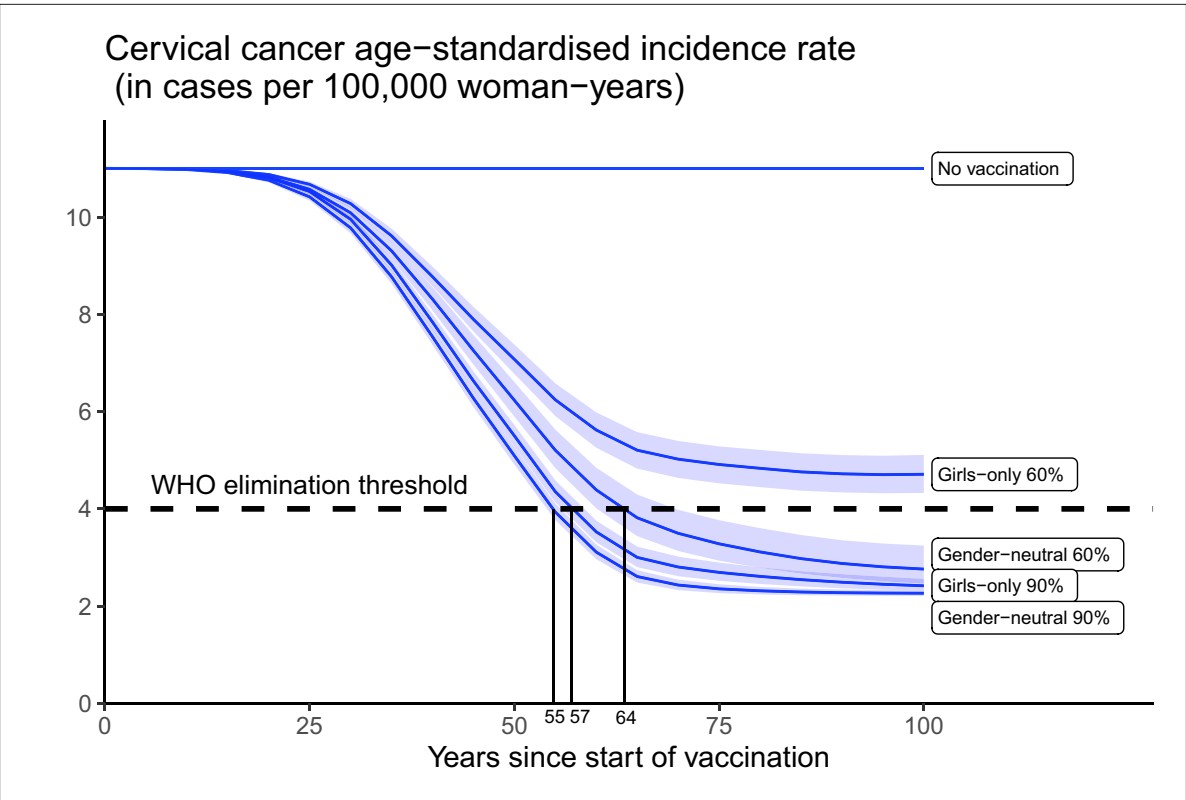

**Figure 3.** Progress towards cervical cancer elimination over time. Predicted cervical cancer age-standardised incidence (in cases per 100,000 woman-years) in the years since start of vaccination in India under no vaccination and in the four highlighted scenarios. The dashed line represents the World Health Organization (WHO) elimination threshold for cervical cancer elimination as a public health priority, that is, age-standardised incidence of 4 cases per 100,000 woman-years. Disruption was simulated according to the base case with a period of disruption of 5 years and 0% coverage in girls and boys during the disruption period.

have the lowest impact, reducing the nationwide ASIR of cervical cancer from 11 cases, which is the present level before the introduction of vaccination, to 4.7 (UI: 4.3, 5.1) cases per 100,000 woman-years in 100 years, but this remained above the WHO elimination threshold (*Figure 3*). The other highlighted scenarios enabled elimination to be reached within 55–62 years. Introducing GN vaccination with 60% coverage in both girls and boys would reduce incidence down to 2.8 (UI: 2.5, 3.2) cases per 100,000 woman-years, hence below the elimination threshold. The GO strategy with 90% coverage, as recommended by WHO, would also decrease incidence below the threshold to 2.4 (UI: 2.3, 2.6) cases per 100,000 woman-years. Lastly, given the already high vaccination coverage of 90% in girls, switching to the GN strategy would only marginally reduce incidence to 2.3 (UI: 2.2, 2.3) cases per 100,000 woman-years.

Other combinations of coverage in boys and girls would also allow elimination. Without vaccination in boys, the WHO elimination threshold could be reached through a critical GO coverage of 70% (UI: 65, 73) (*Figure 4*). With vaccination in boys, a lower critical coverage in girls might be sufficient for elimination. For instance, coverage between 50% and 70% in girls might also be sufficient for elimination when combined with moderate coverage (30%) in boys (*Figure 4*).

Finally, while disruption of vaccination does not prevent countries from reaching the elimination threshold eventually (provided vaccination resumes after disruption), it can prolong the number of years taken to attain this goal (*Appendix 2—figure 4*). Under the GO strategy with 90% coverage, for instance, disruption of vaccination for 5 years (as in the base case) would delay reaching elimination from 57 to 65 years since the start of vaccination. By switching to GN strategy (under 90% coverage), elimination could be reached in 58 years since the start of vaccination, even with disruption.

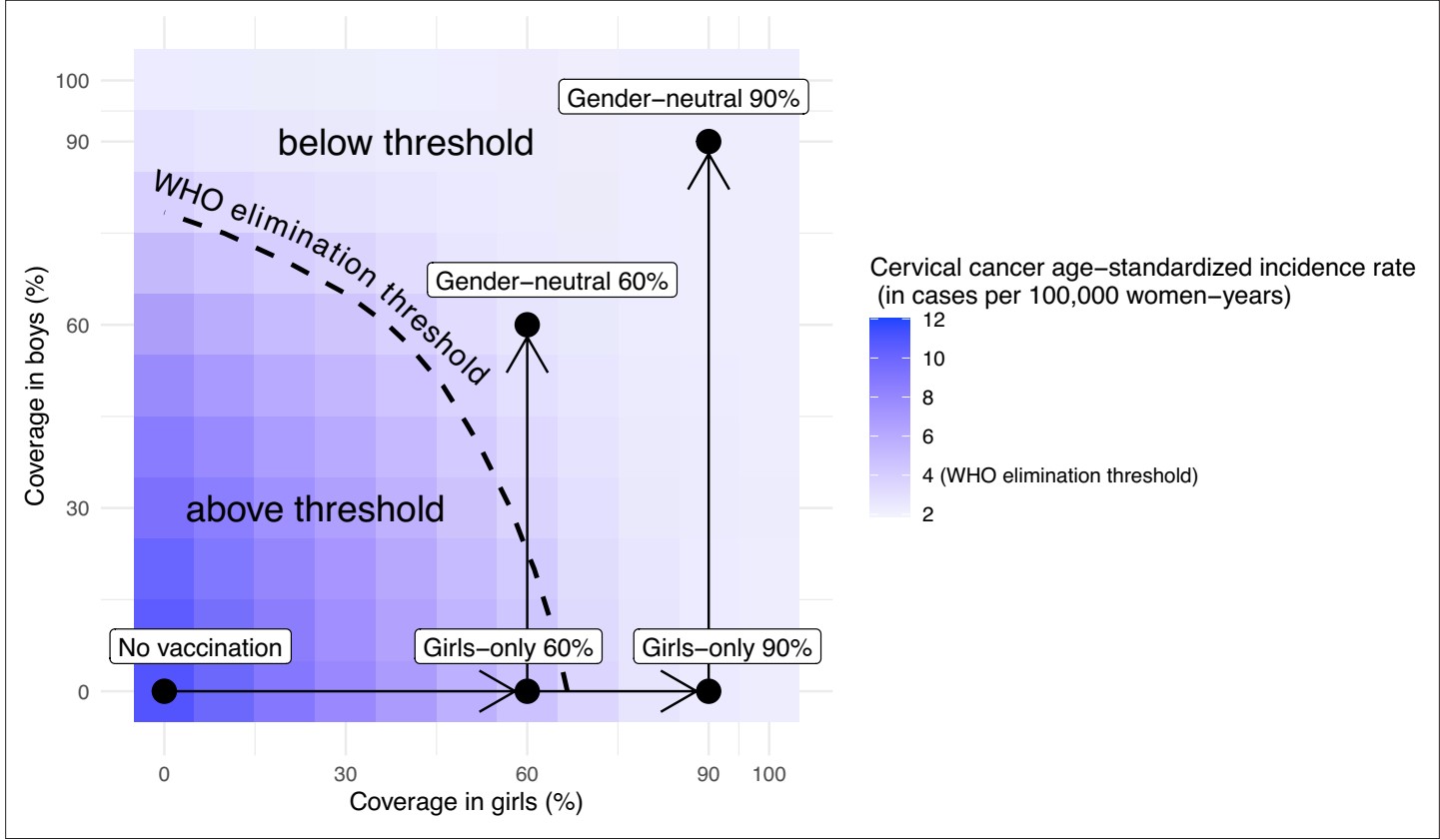

**Figure 4.** Attainment of cervical cancer elimination in the long term. Heatmap of the predicted cervical cancer age-standardised incidence rate (in cases per 100,000 woman-years) in the long term (i.e., at 100 years after the start of vaccination) in India by vaccination coverage in girls and boys (under no disruption of vaccination). The dashed curve represents the World Health Organization (WHO) elimination threshold for cervical cancer elimination, that is, age-standardised incidence of 4 cases per 100,000 woman-years. The five black circles correspond to no vaccination and the four highlighted scenarios: girls-only strategy with 60% coverage, girls-only strategy with 90% coverage, gender-neutral strategy with 60% coverage, and gender-neutral strategy with 90% coverage. The horizontal arrows represent scale-up of vaccination by increasing coverage in girls, and the vertical arrows represent switching to gender-neutral vaccination.

## Discussion

As observed worldwide during the COVID-19 pandemic, public health programmes can be dramatically impacted by the occurrence of sudden societal and infrastructural crises (*WHO, 2021*). The repercussions of disruption of health-care systems may subsequently translate into increased morbidity and mortality (*Sharpless, 2020*), and economic burden (*Richards et al., 2020*), at a population level. In the present paper, using India as an example, we illustrate how the introduction of GN HPV vaccination could mitigate the negative impact of a sudden HPV vaccine delivery disruption. More specifically, using a validated agent-based model, calibrated to context-specific data, we have simulated a range of plausible scenarios of disruption of a national HPV vaccination programme and assessed the gain in resilience by shifting from GO to GN vaccination in terms of the number of cervical cancer cases still prevented in the birth cohort with vaccination disruption.

Our model-based exercise showed that in the case of HPV vaccination disruption for 5 years (base case), resilience increased by 2.8-fold (from 107 to 302 cases per 100,000 girls born) and 2.2-fold (from 209 to 464 cases per 100,000 girls born) when shifting from GO to GN vaccination under 60% and 90% coverage, respectively. GN vaccination was shown to improve resilience irrespective of the duration of disruption. However, the absolute magnitude of resilience and of resilience gain resulting from adding boys to routine vaccination of girls steadily decreased with increasing duration of disruption, under all the vaccination strategies assessed, indicating that measures to restore HPV vaccination should be implemented as soon as feasible.

An important determinant underlying the gain in resilience by switching from GO to GN vaccination is likely the age difference between sexual partners. In general, men are on average older than women within sexual partnerships almost everywhere worldwide (*Wellings et al., 2006*). In cases of vaccination disruption, the birth cohorts of boys vaccinated before the disruption would indirectly protect the cohorts of younger girls who missed out on vaccination during the disruption period. Since sexual behaviour is regulated by population-specific social norms, the magnitude of the resilience attributable to GN HPV vaccination is also expected to be different across populations. In India, for example, the age difference between male and female partners is on average 7 years (*USAID, 2017*), similar to the number of birth cohorts still retaining some protection by GN vaccination when we simulated no restart of vaccination after disruption. As shown by our results, increasing coverage in girls also helps to increase resilience. However, vaccinating boys is a more direct way to reduce the force of infection in men than vaccinating girls, which is the likely mechanism underlying the higher resilience achieved by GN strategy than GO strategy.

In the present paper, we have also illustrated how the introduction of GN vaccination across India would enhance progress towards the elimination of cervical cancer. In principle, elimination across India could be reached with a critical level of GO coverage of 70%, which falls between the average coverage observed among girls aged 9–15 years living in countries with an active HPV vaccination programme, that is, 53% (*Bruni et al., 2021*), and the coverage recommended by the WHO, that is, 90% (*WHO, 2020b*). On the other hand, our model-based estimates suggest that cervical cancer elimination could be reached with 50–70% coverage in girls when combined with moderate coverage in boys, even if coverage in boys were lower than that of girls. In cases where GO coverage is already above 70%, GN vaccination would allow even more ambitious cervical cancer control targets to be reached, which might eventually make cervical screening targeted to vaccinated birth cohorts redundant (*Tota et al., 2020*). Moreover, as previously reported (*Man et al., 2022*), reaching elimination might be challenging in the Indian states with the highest baseline cervical cancer incidence before vaccination. In these cases, additional measures such as GN vaccination would be even more relevant.

The availability and limitations of data on sexual behaviour and cervical cancer incidence across Indian states were discussed in detail in our previous publication in which we presented the calibrated model for the first time (*Man et al., 2022*). A limitation of this study is that the critical levels of HPV vaccination coverage found for India are not necessarily generalisable to other populations. Baseline cervical cancer incidence and the magnitude of herd effect, that is, the indirect protection offered by the vaccinated to the unvaccinated individuals of a population against HPV infection (*Malagón et al., 2018*) should be considered. These are, in turn, governed by local sexual behaviour (*Bosch et al., 1994*; *Schulte-Frohlinde et al., 2022*). As mentioned above for resilience, the critical level of vaccination coverage to reach cervical cancer elimination will be context-specific (*Baussano et al., 2018*; *Lehtinen et al., 2022*). Extrapolation of the results of this study to other populations will be limited to those sharing similar patterns of demography, social norms, and cervical cancer epidemiology as India. Nevertheless, we expect the principle of improving the resilience of cervical cancer prevention through a shift to GN vaccination to be widely applicable, and this is supported by the consistency of our results in India when stratifying by state-specific cervical cancer incidence.

Another limitation is the finite number of scenarios considered in which disruption of a health system might occur. For example, we did not explore changes in sexual behaviour in the population due to disruption, which occurs in dire circumstances like a pandemic. However, we did account for a large variation in the duration of disruption and in vaccination coverage at disruption through our sensitivity analyses. In addition, we did not consider the influence of choice of HPV vaccine, as a quadrivalent vaccine (either indigenously produced or Gardasil) would be the likely choice for India due to the price advantage. In cases of using a nonavalent vaccine, or a vaccine with higher levels of cross-protection, we expect similar qualitative results for resilience, and elimination of cervical cancer could be expected to be more easily achieved. We also did not consider changes in cervical cancer screening, as it is difficult to predict how much coverage of cervical cancer screening will increase in India in the coming years.

Although a formal health-economic assessment of the introduction of GN HPV vaccination in India is beyond the scope of this paper, clearly adding vaccination of boys to the routine coverage of girls would approximately double the required number of vaccine doses and would increase the financial effort to be made by the national government. It has been demonstrated that GN HPV vaccination

schedules are economically attractive in high-income tender-based settings, in particular where GO vaccine uptake is below 80% (*Qendri et al., 2020*). GN vaccination is likely to be even more attractive in India where population-based screening programmes have not yet been widely implemented. Of course, the actual determinants of a trade-off between redundancy and efficiency of resource allocation depends on context-specific planning and assessment. In our example, to achieve greater resilience, GN vaccination with 60% coverage would require more doses of vaccine while leading to fewer prevented cases of cervical cancer in vaccinated cohorts without interruption than GO vaccination with 90% coverage (*Figure 2*). However, extending vaccination to boys could be easier to put in place than improving coverage in girls from 60% to 90% in some contexts. Finally, shifting from a two-dose to a single-dose HPV vaccination schedule could also improve the affordability of GN vaccination. More research is needed to investigate the cost-effectiveness and affordability of GN vaccination in low-resource settings and how resilience should be accounted for in deciding the vaccination strategy.

Having clearly illustrated the merits of GN vaccination considering different aspects across different settings here and in earlier studies (*Chow et al., 2021*; *Elfström et al., 2016*; *Lehtinen et al., 2022*; *Lehtinen et al., 2018*; *Qendri et al., 2020*), the next step is to establish a feasible pathway for implementation (*de Sanjose and Bruni, 2020*). In high-income countries, where affordability was less of an issue, most vaccination programmes were initiated with the GO strategy, often combined with catch-up in girls/women, and are now increasingly shifting towards GN vaccination. In resource-limited settings, once routine vaccination in girls has been implemented, countries may need to make a choice between boys' vaccination or female catch-up vaccination. The limited global vaccine supply will likely remain an important factor influencing this decision-making process in the coming years (*WHO, 2022b*). In addition, the choice at the country level should also account for the health-economic objectives, the type of health infrastructure already in place as well as socio-cultural context. In some contexts, important arguments for choosing GN strategy could be the fact that it helps to overcome the stigma associated with a vaccine that targets only girls and aims to prevent a sexually transmitted disease (*Cernasev et al., 2023*), and the direct health benefits in men who have sex with men.

In conclusion, within the limitations of our model, the model-based estimates show that shifting from GO to GN vaccination may improve the resilience of the Indian HPV vaccination programme while also enhancing progress towards the elimination of cervical cancer. Our resilience estimates go beyond health-care disruption due to the COVID-19 pandemic. Indeed, any societal events undermining the routine activities of a health-care system can produce a comparable effect. Over the years, disease prevention programmes have been disrupted by the occurrence of pandemics (*Osterholm, 2017*; *WHO, 2021*), armed conflicts (*Jawad et al., 2021*; *McGrath, 2022*), economic sanctions (*Germani et al., 2022*), or widespread vaccine hesitancy (*Larson, 2020*). Therefore, we argue that such societal crises, which are unpredictable but expected to occur, should be anticipated through careful planning of disease prevention programmes.

## Materials and methods
### Model

EpiMetHeos was used to simulate a dynamic sexual contact network, which allows long-term and overlapping partnerships, through which HPV infections were transmitted. Based on the impact of vaccination on high-risk HPV incidence derived from EpiMetHeos, Atlas was then used to derive the impact on cervical cancer burden.

The models were previously calibrated to sexual behaviour, HPV prevalence, and/or (delete as appropriate) cervical cancer incidence in India to assess the impact of single-dose vaccination in India, while accounting for the uncertainty in the duration of vaccine protection (*Man et al., 2022*). Since the high-quality data on HPV prevalence and cervical cancer incidence data needed to calibrate the models to each Indian state were not available, we used a Footprinting framework to approximate the missing data and extrapolate the impact by state. In short, we first identified clusters of states with similar patterns of cervical cancer incidence. We then calibrated EpiMetHeos to the state of Tamil Nadu to represent the states in the high cancer incidence cluster, and the state of West Bengal to represent the states in the low cancer incidence cluster, using the available sexual behaviour and HPV prevalence data of these two states. Finally, we estimated the impacts for Tamil Nadu and West Bengal and

extrapolated these to other states within each cluster. The model was calibrated for previous publications (*Man et al., 2022*; *Man et al., 2023*). See more details of the model in Appendix 1 sections 'HPV transmission model', 'Model calibration', and 'Computation of the model outcomes'. This study adheres to HPV-FRAME, a quality framework for modelled evaluations of HPV-related cancer control (*Reporting standard 1*) (*Canfell et al., 2019*).

## Simulation scenarios

Recently, marketing authorisation has been awarded to a locally produced quadrivalent HPV vaccine in India based on successful immunobridging between the new vaccine and the existing quadrivalent vaccine (Gardasil, MSD) (*The Lancet Oncology, 2022*). While it is still uncertain which HPV vaccine will be used in the Indian immunisation programme, we simulated vaccination with a quadrivalent vaccine targeting two high-risk types, HPV16 and HPV18 (and two low-risk types, HPV6 and HPV11). The efficacy estimates of the locally produced HPV vaccine were not available to us; hence, we based our model assumptions on the efficacy estimates of the IARC India vaccine trial, which considered Gardasil (*Basu et al., 2021*; *Joshi et al., 2023*). In this trial, no difference was found between the efficacy estimates of the single-dose and two-dose schedules, which were moreover stable after up to 10 years' follow-up. Hence, we based the model on pooled efficacy estimates (95% efficacy for HPV16 and HPV18, 9% cross-protection for HPV 31, HPV33, and HPV45, and 0% efficacy for the remaining high-risk HPV types) and assumed no waning of vaccine immunity over time. In doing so, the modelled efficacy could represent vaccination under either a single-dose or two-dose schedule. Vaccine protection was modelled through an all-or-nothing working mechanism, that is, if the efficacy is X%, then X% of the vaccinated individuals in the model are fully protected and 100-X% fully not protected.

To assess the resilience of different vaccination strategies on cervical cancer prevention, we simulated the above scenarios first without disruption of the HPV vaccination programme, and then with disruption occurring 10 years after the start of vaccination. As the base case, we simulated a disruption period of 5 years with total interruption of HPV vaccine delivery, that is, 0% vaccination coverage in both boys and girls. As sensitivity analyses, we considered less severe disruption with still 20% and 40% vaccination coverage in girls (but still 0% in boys), as well as shorter or longer durations of disruption of 1, 2, and 10 years, and no restart of vaccination. These sensitivity analyses were considered to represent different types of disruption attributable to a COVID-19 pandemic or to any other interference of the HPV vaccination programme.

## Model outcomes

Estimates of the life-time number of cervical cancer cases prevented per 100,000 girls born were derived by cohort. The first 40 birth cohorts following the introduction of vaccination were considered to have estimates for both cohorts before and after the disruption period. The mean number of cases prevented across the first 10 vaccinated birth cohorts was used to evaluate the impact prior to the disruption. As a measure of resilience, we used the mean number of cases still prevented, as a result of previous HPV vaccination, across the birth cohorts with disruption. Considering such a metric of resilience allows us to evaluate whether any given strategy provides a sufficient level of protection retained during disruption as compared to a possible prefixed target level of protection. Estimates of resilience were compared between the highlighted scenarios A–D based on their ratio.

To assess progress towards cervical cancer elimination, we also derived the ASIR of cervical cancer up to 100 years after the start of vaccination. The long-term impact of vaccination was defined as the impact at 100 years after the start of vaccination. The Segi world standard population was used for the standardisation (*Segi, 1960*). Model estimates of ASIR were compared to the threshold for elimination defined by the WHO of 4 cases per 100,000 woman-years (*WHO, 2020b*).

Model outcomes were reported as the mean and the 10th and 90th percentiles, that is, uncertainty interval (UI), of the simulations using the 100 parameter sets best fitting the sexual behaviour and HPV prevalence data obtained through calibration.

Given the extremely low coverage of the existing screening programme in India, we have not considered the impact of screening in our study (*Bruni et al., 2022*).

## Acknowledgements

This study was funded by the Bill & Melinda Gates Foundation (grant number: INV-039876). For the authors identified as personnel of the International Agency for Research on Cancer (IARC) or WHO, the authors alone are responsible for the views expressed in this article and they do not necessarily represent the decisions, policy, or views of the IARC or WHO. The designations used and the presentation of the material in this article do not imply the expression of any opinion whatsoever on the part of WHO and the IARC about the legal status of any country, territory, city, or area, or of its authorities, or concerning the delimitation of its frontiers or boundaries.

## Additional information

### Competing interests

Rengaswamy Sankaranarayanan: Rengaswamy Sankaranarayanan is affiliated with Karkinos Healthcare. The author has no financial interests to declare. The other authors declare that no competing interests exist.

### Funding

| Funder | Grant reference number | Author |
| --- | --- | --- |
| Bill and Melinda Gates Foundation | INV-039876 | Iacopo Baussano<br>Damien Georges<br>Irene Man |

The funders had no role in study design, data collection and interpretation, or the decision to submit the work for publication.

### Author contributions

Irene Man, Conceptualization, Software, Formal analysis, Validation, Visualization, Methodology, Writing - original draft, Writing – review and editing; Damien Georges, Software, Validation, Visualization, Methodology, Writing – review and editing; Rengaswamy Sankaranarayanan, Partha Basu, Supervision, Funding acquisition, Validation, Writing – review and editing; Iacopo Baussano, Conceptualization, Software, Formal analysis, Supervision, Funding acquisition, Visualization, Methodology, Writing - original draft, Writing – review and editing

### Author ORCIDs

Irene Man http://orcid.org/0000-0003-3177-6904
Damien Georges https://orcid.org/0000-0003-2425-7591
Partha Basu http://orcid.org/0000-0003-0124-4050
Iacopo Baussano http://orcid.org/0000-0002-7322-1862

### Decision letter and Author response

Decision letter https://doi.org/10.7554/eLife.85735.sa1
Author response https://doi.org/10.7554/eLife.85735.sa2

## Additional files

### Supplementary files

- MDAR checklist

- Supplementary file 1. List of model parameters.

- Supplementary file 2. Data related to Appendix 1. (**A**) Overview of available cancer incidence data from local registries by Indian state. (**B**) Age-specific cervical cancer incidence data by Indian state. (**C**) Mortality rate of India. (**D**) Type-specific contribution of HPV types in cervical cancer. (**E**) Standard world population (Segi, 1960). (**F**) Female population size by Indian state.(**G**) Pre-vaccination risk of cervical cancer by Indian state.

- Supplementary file 3. Sensitivity analyses on coverage at disruption and duration of disruption on resilience by Indian state.

- Reporting standard 1. HPV-FRAME checklist (*Canfell et al., 2019*).

### Data availability

All data used in the present study were openly available and extracted from http://ci5.iarc.fr for the cervical cancer incidence data published by the International Agency for Research on Cancer, from https://www.ncdirindia.org/All_Reports/Report_2020/resources/NCRP_2020_2012_16.pdf for the cervical cancer incidence data published by the National Centre for Disease Informatics and Research of India, from https://doi.org/10.1038/sj.bjc.6602348 and https://doi.org/10.1097/pgp.0b013e3182399391 for the HPV prevalence data, from https://www.aidsdatahub.org/sites/default/files/resource/national-bss-general-population-india-2006.pdf for the sexual behaviour data published by the National AIDS Control Organisation Ministry of Health and Family Welfare Government of India, from https://doi.org/10.1128/jcm.00354-11 for the data on type-specific contribution of HPV types in cervical cancer, from https://population.un.org/wpp/Download/ for the mortality rates data published by the United Nations, from https://censusindia.gov.in/census.website/data/census-tables for the Indian state-specific census data published by the General Census Commissioner India, from https://cdn.who.int/media/docs/default-source/gho-documents/global-health-estimates/gpe_discussion_paper_series_paper31_2001_age_standardization_rates.pdf for the Segi standard world population data. The computer code regarding the HPV transmission model EpiMetHeos is a part of the METHIS (ModElling Tools for HPV Infection-related cancers) project, which will gradually make publicly available a set of open-source models on a platform at the webpage https://iarc-miarc.gitlab.io/methis/methis.website/about/ from now to 2025. Before 2025, the code of EpiMetHeos can be made available upon request to the authors. Requests will be assessed on a case-by-case basis in consultation with lead investigators and co-investigators. The computer code regarding the cervical cancer progression model Atlas, also a part of METHIS, is already publicly available on the platform webpage.

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

## Appendix 1

### HPV transmission model

We adapted a previously published HPV transmission model (*Baussano et al., 2013*) into a stochastic agent-based dynamic model, EpiMetHeos. EpiMetHeos is an extension of EpiModel (*Jenness et al., 2018*) an open-source statistical framework that allows simulation of infectious disease transmission on dynamic contact networks.

### Demography

The model considers an open population in which individuals enter at the age of 10 years and exit due to death with maximum age of 100 years. Each individual is characterised by the two demographic characteristics, age and sex, denoted by $a \in [10, 100]$ and $g \in \{W, M\}$, respectively. The entire age range is further stratified into the following age groups: $ageg \in \{10-14, 15-19, 20-24, 25-29, 30-49, 50-59, 60-99\}$.

Individuals die according to sex- and age-specific probabilities $m_{g,age}$. The number of new individuals born into the population per time step is given by $b$, which is set to a value that keeps the total population constant over time and with 50%–50% of female and male newborns. A constant age-specific population size distribution corresponding to $m_{g,age}$ is achieved through the simulation of a burn-in period.

### Sexual contact behaviour

Two dynamic contact networks of sexual partnerships are modelled among the model individuals: one for stable partnerships and one for one-off partnerships. Stable partnerships represent marital partnerships and one-off partnerships outside marriage. The networks are constructed using EpiModel's implementation of the Separable Temporal Exponential-family Random Graph Models (STERGM) (*Jenness et al., 2018*). An STERGM is characterised by formation and dissolution probabilities of partnerships. At each time step, partnerships that were present at the previous time step can be dissolved, and new partnership can be formed between individuals that were not connected at the previous time step.

Formation of partnerships is only allowed between opposite sex individuals to model heterosexual networks. Formation probabilities of stable as well as one-off partnerships depend on the individual's sex, age group, and the assigned risk group of sexual activity. Each individual is assigned to one of the five risk groups of sexual activity, denoted by $riskg \in \{1, 2, 3, 4, 5\}$, which determines which type of partnerships he/she is allowed to form as follows:

> $riskg = 1$: no stable, no one-off,
> $riskg = 2$: only stable,
> $riskg = 3$: both stable and one-off,
> $riskg = 4$: only one-off,
> $riskg = 5$: only one-off; in women, this risk group represents female sex workers.

The risk group of an individual is assigned randomly at birth according to a sex-specific multinomial distribution $p_{g,riskg}$ and remains unchanged for the rest of the individual's lifespan. The dissolution probability differs between stable and one-off partnerships but is the same for all individuals.

Sexual behaviour is further characterised based on parameters regarding the number of sex acts within established partnerships. At each time step, sex acts may occur within a stable partnership. The number of sex acts is randomly generated according to a Poisson distribution with mean $r^{main}$. Within a one-off partnership, the number of sex acts $r^{one-off}$ is exactly one. The parameter values regarding sexual contact behaviour were fixed or obtained through a calibration step described in Section 'Calibration to sexual behaviour data'.

### HPV natural history

HR HPV types are assumed to be transmitted independently, governed by type-specific natural history parameters, including the probability of transmission, duration of infection, and natural immunity. Transmission of all HR HPV types are assumed to follow and the 'Susceptible-Infected-Removed/Immune-Susceptible' dynamics in women (*Appendix 1—figure 1*). In men, it has generally been observed that the HPV prevalence and the rate of acquiring new HPV infection

remain constant across age, and that the seroconversion rate is low after natural infection (*Giuliano et al., 2008*; *Giuliano et al., 2015*; *Schiffman et al., 2016*). Hence, we assumed the 'Susceptible-Infected-Susceptible' dynamics in men. Effects of HIV, smoking, and use of contraceptives, as risk factors of HPV transmission and cervical cancer progression, were not included in the model. We chose not to include them in the model. For HIV infection, this choice was based on the low HIV prevalence in India (*Stelzle et al., 2021*). As for smoking and the use of contraceptives, the choice was based on the lack of data.

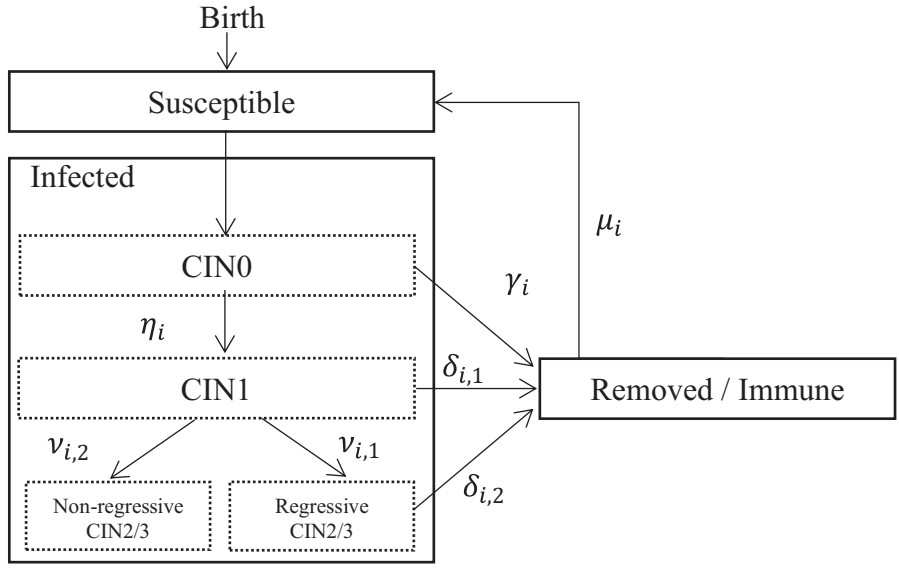

**Appendix 1—figure 1.** Structure of HPV natural history in EpiMetHeos.

All individuals enter the population being susceptible. At each time step, an unvaccinated susceptible individual having $l$ sex acts with an individual infected with type $i$ has a probability of $1 - (1 - \beta_i)^l$ becoming infected, where $\beta_i$ is the probability of transmission per sex act. If vaccinated and vaccine protection is successfully induced, the probability of transmission for a vaccine-targeted type, either a vaccine type or a cross-protective type, becomes zero. In other words, we assumed an all-or-nothing working mechanism for vaccine protection. When vaccine-induced protection is lost, the probability of transmission returns to $1 - (1 - \beta_i)^l$.

The duration of infection follows a type-specific distribution with six parameters $\gamma_i$, $\eta_i$, $\delta_{i,1}$, $\delta_{i,2}$, $\nu_{i,1}$, $\nu_{i,2}$ . This distribution describes the process through CIN0, CIN1, regressive CIN2/3, and non-regressive CIN2/3 stages as modelled in an extensively validated cervical cancer progression model (*Berkhof et al., 2013*; *Bogaards et al., 2010*). In particular, non-regressive CIN2/3 here represents the part of HPV infections that will persist and progress to cancer. For those HPV infections that do not persist, upon clearance, men become susceptible and women become removed/immune. Note that while the clearance distribution of HPV infection in the model was constructed by reproducing the transitions through different intermediate CIN stages, we do not explicitly simulating the prevalence or incidence of these intermediate stages to reduce the computational load of simulation. The duration of natural immunity in women follows an exponential distribution type-specific rate $\mu_i$ , after which women also become susceptible again. The values of the natural history parameters were fixed or obtained through a calibration step described in the section 'Calibration to HPV prevalence data' and are reported in *Supplementary file 1*.

## Model calibration

### Footprinting framework
Since high-quality data about HPV prevalence and cervical cancer incidence data essential to project the impact of HPV vaccination are not available for each Indian state, the Footprinting framework was used to approximate missing data and extrapolate impact projections. The Footprinting framework is described in more details in a separate manuscript (*Man et al., 2023*). The framework

consists of three steps: clustering, classification, and projection. These steps are briefly described below.

## Clustering step

In the clustering step, Indian states with available cervical cancer incidence data were clustered based on their similarity in patterns of age-specific cervical cancer incidence. Sources for cervical cancer incidence data were volume XI of Cancer Incidence in Five Continent (CI5) and the 2012–2016 Report of National Cancer Registry Programme by National Centre for Disease Informatics and Research (NCDIR) (*Bray et al., 2017*; *National Centre for Disease Informatics and Research, 2020*). See *Supplementary file 2A* for an overview of Indian states (or groups of states) with available cancer incidence data from local registries. Of the 25 Indian states, 14 states have local registries and 11 do not. Whenever a registry was both present in CI5 and NCDIR, only the data by CI5 were used. See *Supplementary file 2B* and *Appendix 1—figure 4* for the extracted age-specific cervical cancer incidence data.

A Poisson-regression-based CEM clustering algorithm was used to cluster the age-specific cervical cancer incidence curve. Details of the clustering algorithm described in Appendix 1 of the separate manuscript are reported in the main text of the separate manuscript (*Man et al., 2023*). We identified one cluster with high cancer incidence and one with low cancer incidence. See also column 'cluster' of *Supplementary file 2B* for the assigned cluster of each Indian state. In this table, the Indian states with cervical cancer incidence, hence involved in this clustering step, are indicated by 'extracted' in the column 'source'.

## Classification step

The remaining Indian states without cervical cancer incidence data were classified to the identified clusters of Indian states with similar patterns of cervical cancer incidence based on similarity in sexual behaviour data from the behaviour surveillance survey by the National AIDS Control Organization of India (*National AIDS Control Organisation, 2006*).

Random forest was used for classification. Details of the classification method are reported in the main text of the separate manuscript (*Man et al., 2023*). See also column 'cluster' of *Supplementary file 2B* for the assigned cluster of each Indian state. In this table, the Indian states without cervical cancer incidence, hence involved in this classification step, are indicated by 'inferred' in the column 'source'.

## Projection step

In the projection step, baseline (i.e., in the scenario without vaccination) HPV prevalence and cervical cancer incidence were approximated, based on the available data within each cluster. For HPV prevalence, high-quality type- and age-specific HPV prevalence data were only available for Tamil Nadu and West Bengal, the former being in the 'high' and the latter in the 'low' incidence cluster (*Dutta et al., 2012*; *Franceschi et al., 2005*). As for cervical cancer incidence data, approximation was based on the mean age-specific incidence within each cluster. See *Appendix 1—figure 5* and *Supplementary file 2B* for mean values. From the approximated Indian state-specific baseline cervical cancer incidence, we then derived cervical cancer risk in terms of three indicators, life-time risk (LTR), and ASIR, according to the methodology described in the sections LTR of cervical cancer and ASIR of cervical cancer, respectively. See *Supplementary file 2G* for baseline cervical cancer risk by LTR and ASIR.

HPV incidence and cervical cancer risk under vaccination scenarios were obtained as follows. The HPV transmission model, EpiMetHeos, was calibrated to these two representative Indian states (Tamil Nadu and West Bengal) according to the procedure described in the section Calibration to sexual behaviour data. Using the two obtained models, we then simulated different vaccination scenarios and obtained estimates of relative reduction in HPV infection risk according to the methodology described in the section Cumulative risk of HPV infection. Finally, these estimates of relative reduction were applied to the previously obtained baseline values of cervical cancer risk to derive the values under vaccination scenarios.

## Calibration to sexual behaviour data

The model was calibrated to the two representation states: West Bengal and Tamil Nadu. Model calibration consists of two steps. In the first step, the component of the model simulating the dynamic

contact networks was calibrated to the sexual behaviour data without considering the component concerning the natural history of HPV yet. Four sources of sexual behaviour data were used:

- The Demographic and Health Survey (DHS) programme, to provide information on stable partnerships in West Bengal and Tamil Nadu (**USAID, 2017**).
- National Behavioural Surveillance Survey: General Population 2006 by The National AIDS Control Organisation (NACO), to provide information on one-off partnerships in West Bengal and Tamil Nadu (**National AIDS Control Organisation, 2006**).
- Publication by Gaffey et al., 'Male use of female sex work in India: a nationally representative behavioural survey', to provide information on proportion of men with stable partnership that also have one-off partnerships in West Bengal and Tamil Nadu (**Gaffey et al., 2011**).
- Publication by Vandepitte et al., 'Estimates of the number of female sex workers in different regions of the world', to provide information on one-off partnerships across India (**Vandepitte et al., 2006**).

First, the probabilities of being assigned to different risk groups of sexual activity were fixed. See **Supplementary file 1** for the fixed values and the justification. Subsequently, formation and dissolution probabilities of the dynamic sexual contact networks were fitted to a set of target statistics using the *netest* function of EpiModel. The set of target statistics was:

- Stable partnerships:
  - Sex- and age-group-specific population proportion with stable partnerships $d_{g,ageg}^{stable}$
  - Sex- and risk-group-specific population proportion with stable partnerships $d_{g,riskg}^{stable}$
  - Overall population proportion with stable partnerships $d^{stable}$
  - Mean age difference in stable partnerships $\kappa^{stable}$
  - Mean spread of absolute age difference in stable partnerships $\omega^{main}$
  - Mean duration of stable partnerships $s^{stable}$
- One-off partnerships:
  - Sex- and age-group-specific mean degree of one-off partnerships $d_{g,ageg}^{one-off}$
  - Sex- and risk-group-specific mean degree of one-off partnerships $d_{g,riskg}^{one-off}$
  - Overall mean degree of one-off partnerships $d^{one-off}$
  - Mean duration of one-off partnerships $s^{one-off}$

See **Supplementary file 1** for the values of the target statistics and the corresponding data sources used. See **Supplementary file 1** also for the fixed parameters and the corresponding data sources used. See **Appendix 1—figure 2** for the fit to the target statistics. Note that assortativeness of sexual contact by age and risk groups are results of the fitting process.

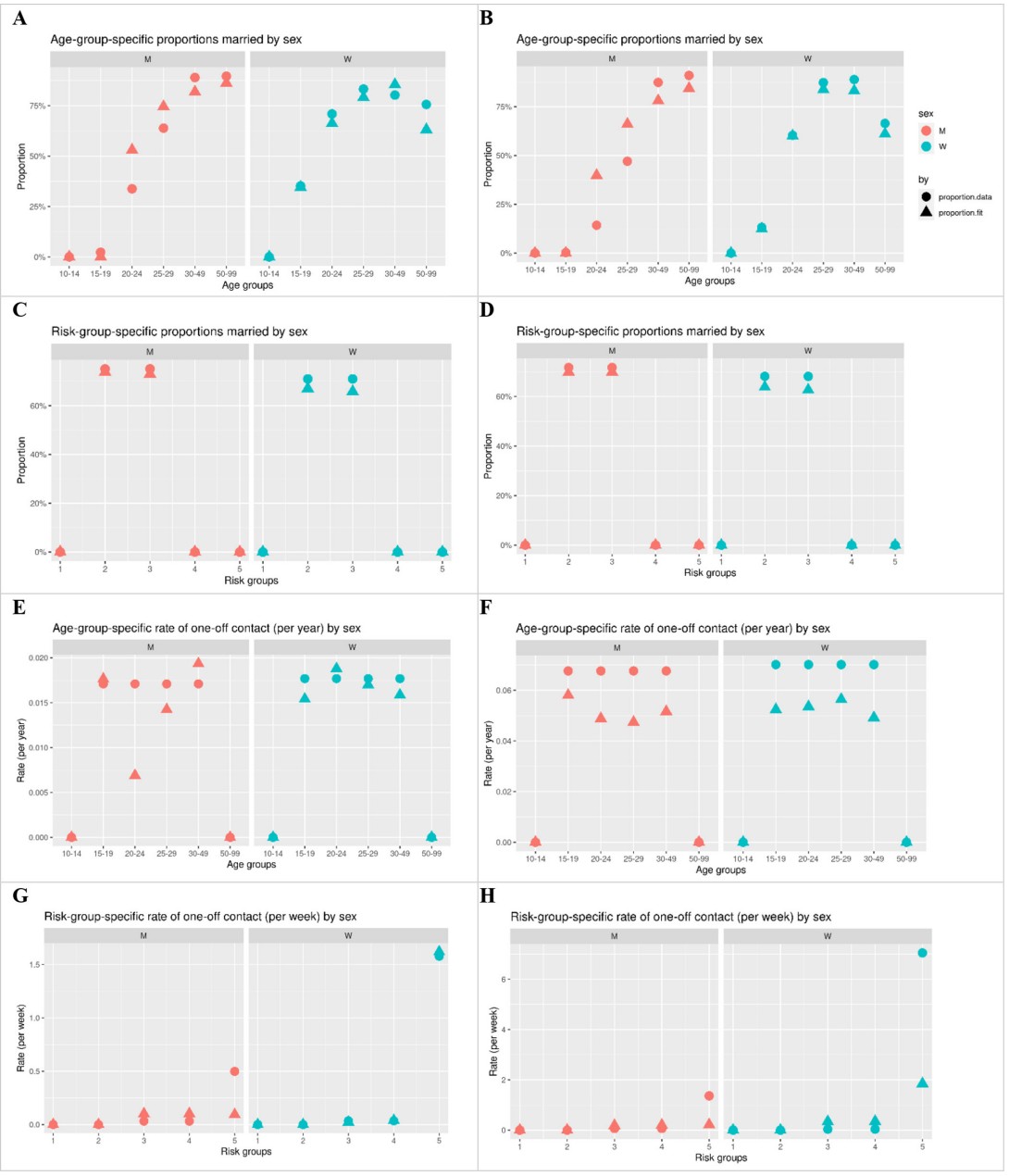

**Appendix 1—figure 2.** Model fit of the target statistics of sexual contact behaviour. Left column: West Bengal. Right column: Tamil Nadu. Circle: targets derived from data. Triangle: fit by model.

## Calibration to HPV prevalence data

In the second step, the entire model was calibrated, including the component concerning the natural history of HPV. This was done by fitting the model HPV prevalence among sexually active women to HPV prevalence data from two surveys among married women in Tamil Nadu and West Bengal (*Dutta et al., 2012*; *Franceschi et al., 2005*). For the study in Tamil Nadu (*Franceschi et al., 2005*), we could extract the age-specific prevalence data for all HR HPV types from the database at IARC. For the study in West Bengal (*Dutta et al., 2012*), we had only the data reported in the publication, which include the age-specific prevalence of HPV 16 and 18 but not that of other HR HPV types. To derive the prevalence of other HR HPV types in West Bengal and ensure consistent type-specific HPV natural history parameters between Tamil Nadu and West Bengal, we scaled the relative type-specific prevalence of Tamil Nadu to West Bengal. The relative prevalence of HPV 16/18 between Tamil Nadu and West Bengal was used as the scaling factor.

Due to the low prevalence of some HR HPV types, we used the average prevalence of the following three groups of HR HPV types as target prevalence:

- HPV 16
- HPV 18
- HPV 31/33/45/35/39/51/52/56/58/59/68

Note that for the subsequent simulation of the vaccination scenarios, the calibrated parameters based on average prevalence of HPV 31/33/45/35/39/51/52/56/58/59/68 were then used to model the cross-protective types HPV 31/33/45 and the other HR HPV types HPV 35/39/51/52/56/58/59/68, which are referred to as '*cross*' and '*other*'.

The parameter values regarding the type-specific progression, clearance rates, and waning rates of natural immunity were fixed to those estimated for the extensively validated cervical cancer progression model (*Berkhof et al., 2013*; *Bogaards et al., 2010*). For HPV 31/33/45/35/39/51/52/56/58/59/68, we derived the mean values of the type-specific estimates.

The parameter values obtained in this calibration step regard type-specific transmission probabilities and one-off partnership underreporting rate. Using 2500 parameter sets that were uniformly generated from the range (*Supplementary file 1*), we selected 100 best-fitting parameter sets. Model fit was evaluated based on log-likelihood of the observed HPV prevalence data given the simulated HPV prevalence under a binomial distribution. For each parameter set, log-likelihood was computed at each year in the last 50 years of 250 years of simulation. The maximum and mean log-likelihood across these years were used as the two summary statistics. See *Appendix 1—figure 3* for the fit to the HPV target prevalence.

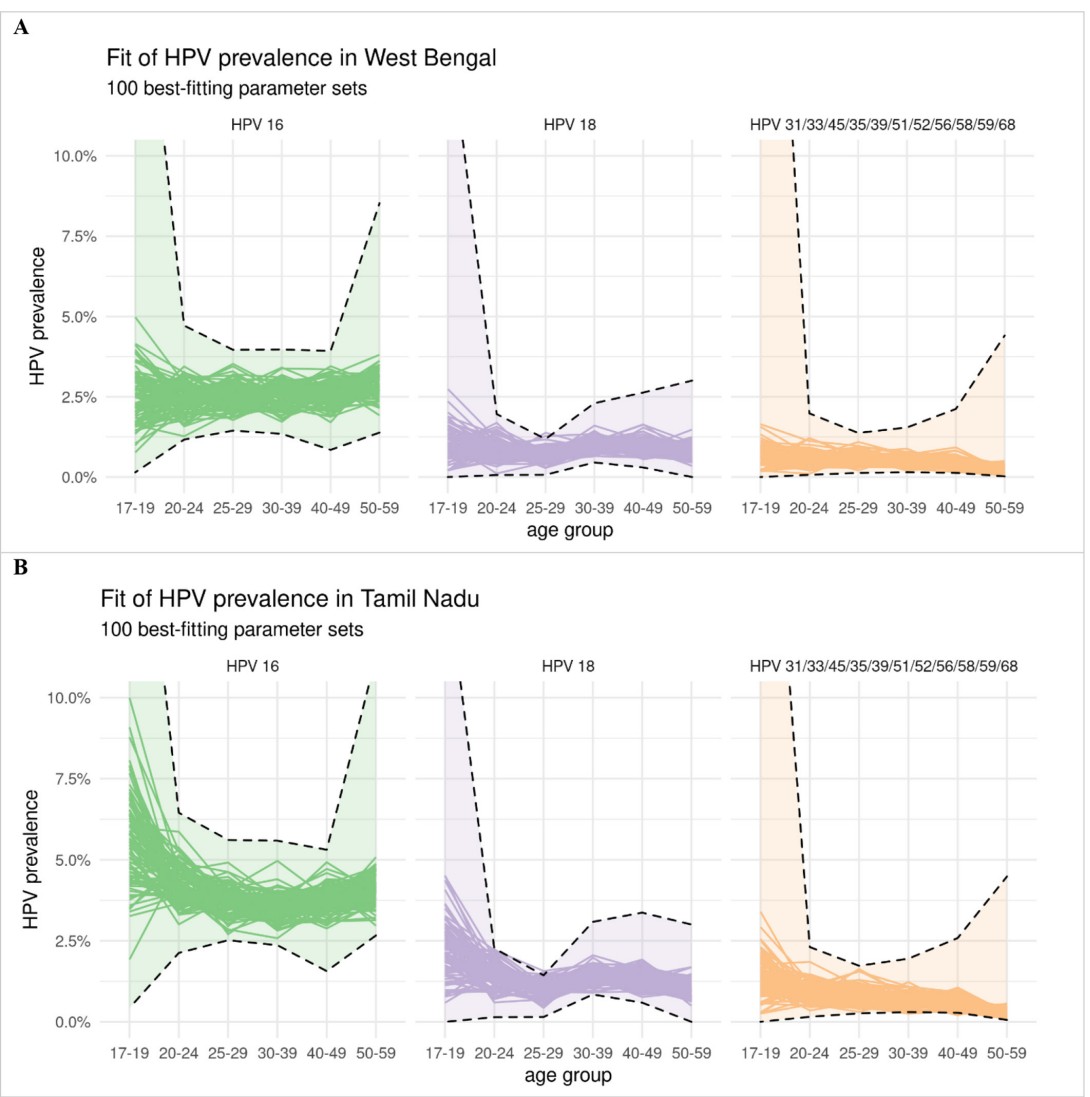

**Appendix 1—figure 3.** Model fit of the type-specific HPV prevalence data. Panel A: West Bengal. Panel B: Tamil Nadu. Model estimates of HPV prevalence of each of the 100 best-fitting parameter sets are given by a separate line. The confidence intervals of the observed HPV prevalence under binomial distribution are given by the dashed lines. The confidence intervals for West Bengal follow the same shape as those for Tamil Nadu, as the type-specific prevalence for West Bengal was derived by rescaling that of Tamil Nadu (the prevalence for West Bengal being approximately half that of Tamil Nadu).

## Computation of model outcomes

### Cumulative risk of HPV infection

For each of the four representative HPV type $i$ and each birth cohort, model estimates on half-year incidence $r_{i,a}$ up to age $a = 40$ years were combined to derive the cumulative risk of HPV infection $Y_i$, as follows: $Y_i = \sum_{a=15}^{45} 0.5 r_{i,a} s_a$. Here, $s_a = \exp\left(-\sum_{a'=10}^{4} 0.5 d_{a'}\right)$ is the survival probability up to age $a$ derived from the mortality rates $d_{a'}$ of the UN life table of India in 2015–2020 (**Supplementary file 2C**).

The cumulative risk of any HR HPV infection was a weighted average of the relative reduction in type-specific cumulative risk of HPV infection. The weights were based on the type-specific contribution to cervical cancer as observed in India (**Basu et al., 2011**). The contributions by HPV 16, 18, and 31/33/45 were 62%, 16%, and 6%, respectively (**Supplementary file 2D**). These proportions were derived by normalising the type-specific contributions.

As for outcome of HPV prevalence, we assumed the model estimates for cumulative risk of HPV infection for West Bengal and Tamil Nadu to apply to all states in the low- and high-cancer-incidence clusters, respectively.

## LTR of cervical cancer

To obtain the LTR of cervical cancer in the scenario without vaccination, we extracted age-specific cervical cancer incidence data from volume XI of Cancer Incidence in Five Continent (CI5) and the 2012–2016 Report of National Cancer Registry Programme by National Centre for Disease Informatics and Research (NCDIR) (*Bray et al., 2017*; *National Centre for Disease Informatics and Research, 2020*) Of the 25 Indian states, 14 states have local registries and 11 do not. Whenever a registry was present in both CI5 and NCDIR, only the data corresponding to CI5 were included. As described in the section Footprinting framework, each state without a local cancer registry was classified to either the cluster of states with high or low cervical cancer incidence. The missing incidence were inferred based on the cluster mean of the classified cluster. See *Supplementary file 2B*, *Appendix 1—figure 4*, and *Appendix 1—figure 5* for the extracted or inferred age-specific cervical cancer incidence data by Indian state.

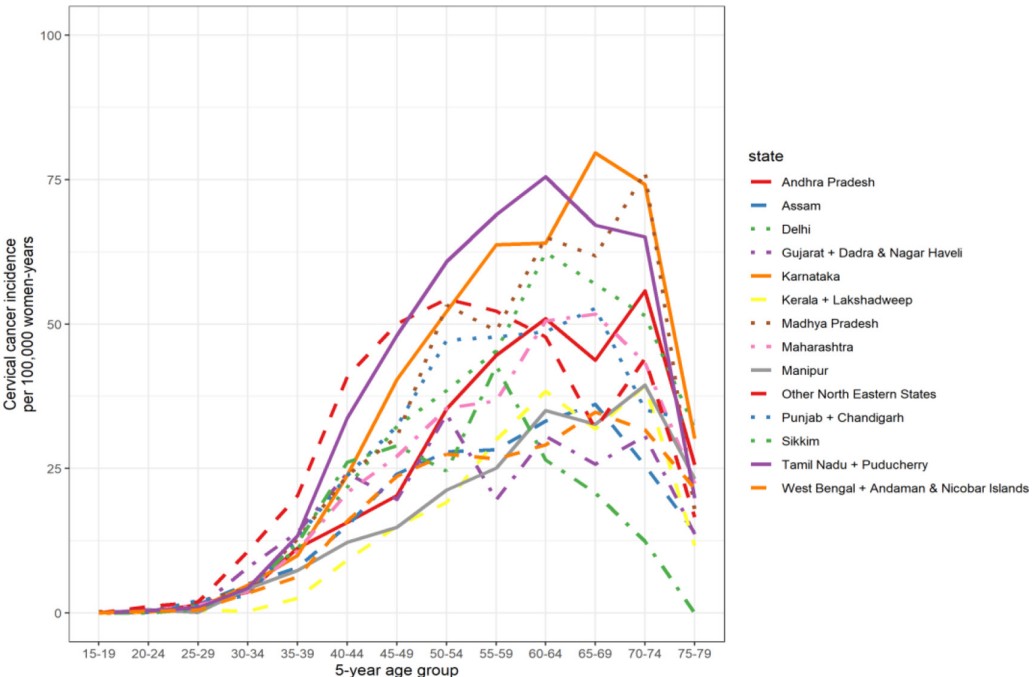

**Appendix 1—figure 4.** Age-specific cervical cancer incidence data by Indian state.

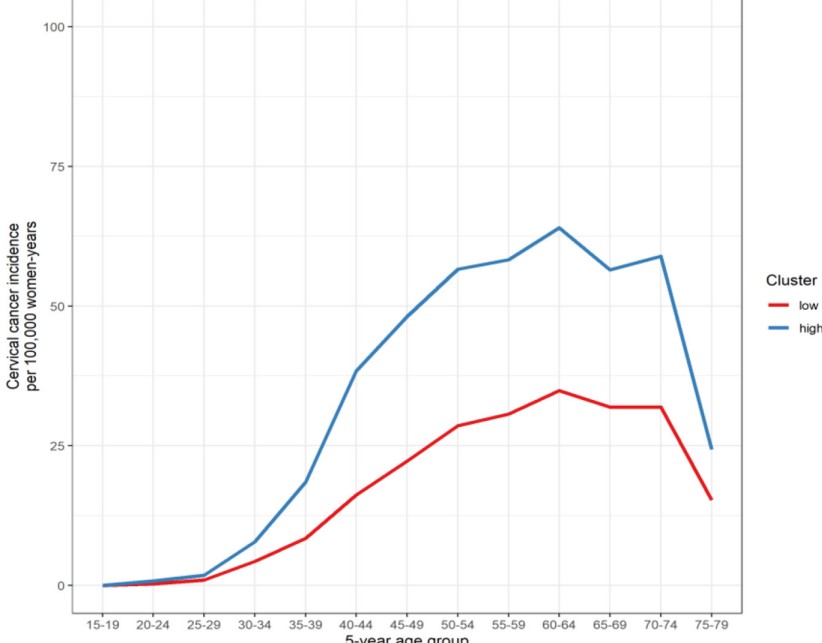

**Appendix 1—figure 5.** Mean of age-specific cervical cancer incidence of the cluster of Indian states.

A recently published method (*Bonjour et al., 2021*) was used to derive LTR of cervical cancer from age-specific incidence. This method converts age-specific cancer incidence into LTR of cervical cancer (in cases per 100,000 girls born) while accounting for the competing risk of dying from other causes before possible occurrence of cervical cancer. For LTR, we used the UN data on female mortality rates for 2015–2020 in India (*Supplementary file 2C*, *United Nations Department of Economic Social Affairs Population Dynamics, 2022*). See *Supplementary file 2G* for the baseline LTR without vaccination by Indian state.

To derive the LTR of cervical cancer in the scenarios with vaccination, we approximated the relative reduction in risk of cervical cancer by model estimates of the relative reduction in the cumulative risk of any HR HPV infection. Model estimates of West Bengal and Tamil Nadu were used for the states in the low- and high-cancer-incidence clusters, respectively.

## ASIR of cervical cancer

ASIR of cervical cancer (in cases per 100,000 woman-years) was obtained based on the world standard population (*Supplementary file 2E*, *Segi, 1960*). See *Supplementary file 2G* for the baseline ASIR without vaccination by state. As for the LTR, the ASIR in the scenarios with vaccination was derived based on model estimates of the relative reduction in the cumulative risk of any HR HPV infection.

## Aggregating model outcomes

To obtain outcomes for India as a whole and by low- and high-incidence states, state-specific outcomes were weighted based on the state-specific female population size according to the table C-13 by the Indian Census of 2011 (*Supplementary file 2F*, *Office of the Registrar, General Census Commissioner India, 2011*).

Model outcomes corresponding to the simulation of the 100 best-fitting parameter sets were used to derive the mean and the 10th and 90th percentiles, that is, UI, of the model outcomes. For the nationwide model outcomes, the 100 best-fitting parameter sets of West Bengal and Tamil Nadu were paired for derived 100×100 outcomes, which were subsequently used to derive the mean and the 10th and 90th percentiles. Furthermore, nationwide outcomes and outcomes across all high- and low-cancer-incidence states were obtained by weighting the state-specific outcomes by the corresponding population sizes.

# Appendix 2

Please see *Supplementary file 3*

## Supplementary results in figures

### A. Indian states with high cervical cancer incidence

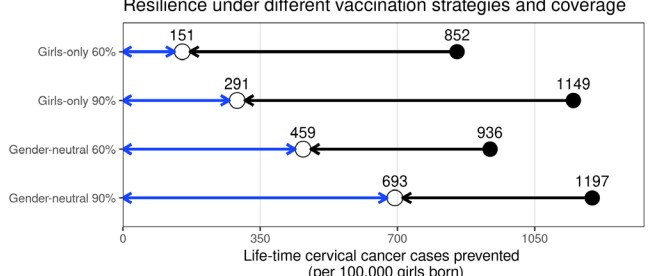

### B. Indian states with low cervical cancer incidence

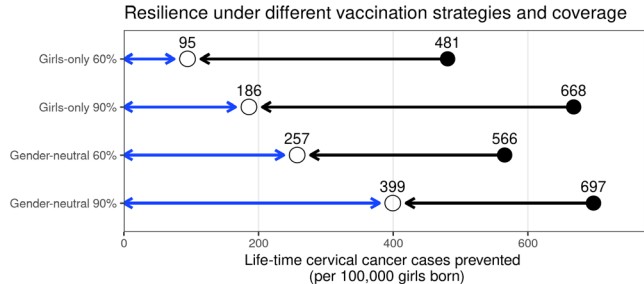

**Appendix 2—figure 1.** Resilience against HPV vaccination disruption in the base case by Indian state. Predicted HPV vaccination resilience, defined as life-time number of cervical cancer cases still prevented in the birth cohorts with disruption of vaccination per 100,000 girls born (blue arrow), and drop in cervical cancers prevented as compared to impact in the undisrupted cohorts (black arrow), under the four highlighted scenarios in Indian states with (**A**) high and (**B**) low cervical cancer incidence. Disruption was simulated according to the base case with a period of disruption of 5 years and 0% coverage in girls and boys during the disruption period. *Figure 1* in the main text corresponds to the results for all Indian states.

### A. All Indian states

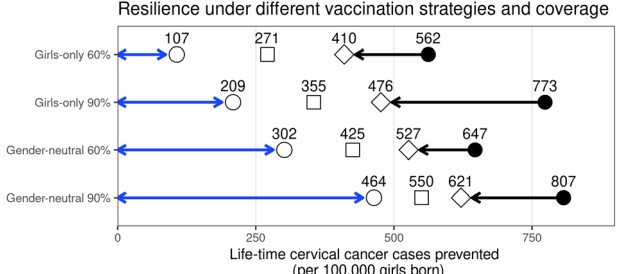

### B. Indian states with high cervical cancer incidence

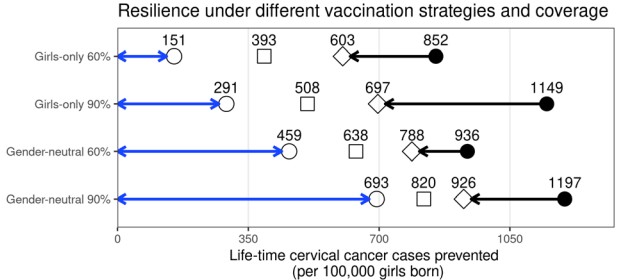

### C. Indian states with low cervical cancer incidence

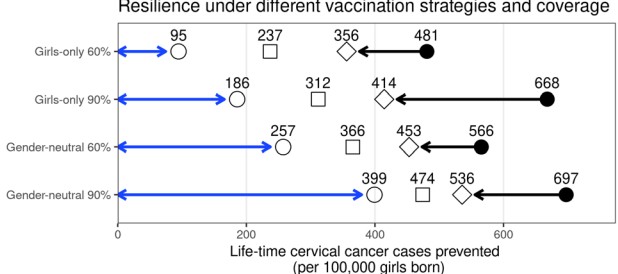

**Appendix 2—figure 2.** Resilience against HPV vaccination disruption in sensitivity analyses on coverage at disruption by Indian state. Predicted HPV vaccination resilience, defined as life-time number of cervical cancer cases still prevented in the birth cohorts with disruption of vaccination per 100,000 girls born (blue arrow), and drop in cervical cancers prevented as compared to impact in the undisrupted cohorts (black arrow), under different vaccination strategies and coverage (rows) in the sensitivity analyses of the coverage of disruption in (**A**) all Indian states, (**B**) Indian states with high cervical cancer incidence, and (**C**) Indian states with low cervical cancer incidence. Disruption was simulated for 0% (base case), 20%, or 40% coverage in girls at disruption. The vaccination coverage was fixed at 0% in boys, and the duration of disruption was fixed at 5 years.

A. All Indian states

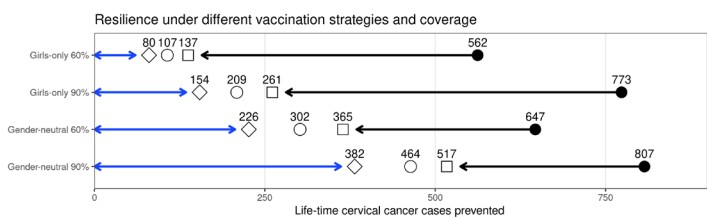

B. Indian states with high cervical cancer incidence

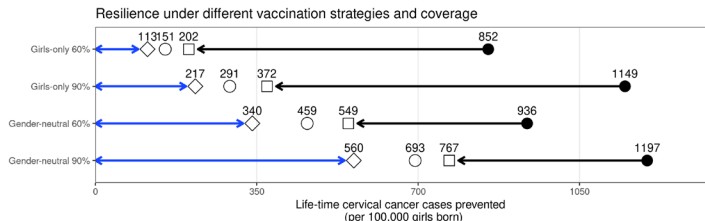

C. Indian states with low cervical cancer incidence

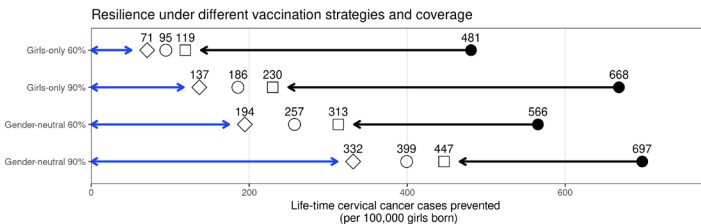

**Appendix 2—figure 3.** Resilience against HPV vaccination disruption in sensitivity analyses on duration of disruption by Indian state. Predicted HPV vaccination resilience, defined as life-time number of cervical cancer cases still prevented in the birth cohorts with disruption of vaccination per 100,000 girls born (blue arrow), and drop in cervical cancers prevented as compared to impact in the undisrupted cohorts (black arrow), under different vaccination strategies and coverage (rows) in the sensitivity analyses of the duration of disruption in (**A**) all Indian states, (**B**) Indian states with high cervical cancer incidence, and (**C**) Indian states with low cervical cancer incidence. Disruption was simulated for 1, 5 (base case), or 10 years indicated by different empty shapes. Vaccination coverage was fixed at 0% in girls and boys during the period of disruption.

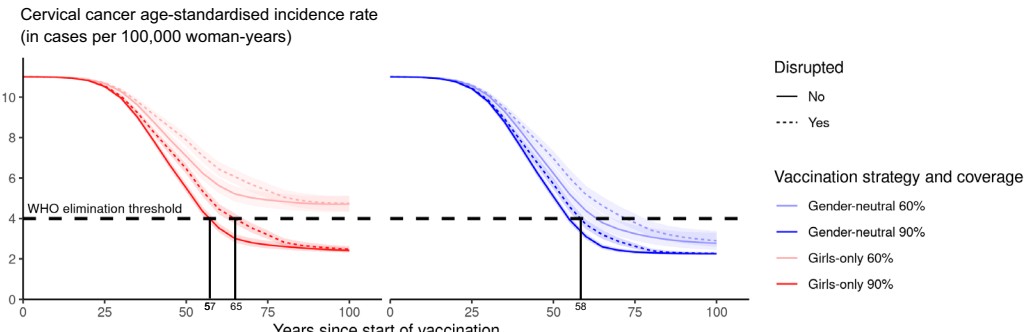

**Appendix 2—figure 4.** Progress towards cervical cancer elimination over time with and without disruption. Predicted cervical cancer age-standardised incidence (in cases per 100,000 woman-years) in the years since start of vaccination in the four highlighted scenarios with (dashed curves) and without (solid curves) disruption. The dashed horizontal line represents the World Health Organization (WHO) elimination threshold for cervical cancer elimination, that is, age-standardised incidence of 4 cases per 100,000 woman-years. Disruption was simulated according to the base case with a period of disruption of 5 years and 0% coverage in girls and boys during the disruption period.

