## [Editor Report]

This study presents valuable findings on how gender-neutral vaccination against human papillomavirus can help improve program resilience in the case of vaccination disruptions. The evidence supporting the claims of the authors is convincing, although the results are only applicable to India and other countries with a similar HPV context; researchers can adapt the model for their local context and use it as a starting point for future research.

---

## [Decision Letter]

**Decision letter after peer review:**

Thank you for submitting your article "Building resilient cervical cancer prevention through gender-neutral HPV vaccination" for consideration by *eLife*. Your article has been reviewed by 3 peer reviewers, and the evaluation has been overseen by a Reviewing Editor and Diane Harper as the Senior Editor. The following individual involved in the review of your submission has agreed to reveal their identity: Romina Tejada (Reviewer #1).

Essential revisions:

– Provide more context on HPV vaccination programs and screening in India.

– Reviewers questioned the calculation methods used for resilience ratios. Consider providing more rationale for these ratios and/or using other metrics for measuring resilience.

– Address requests for clarifications made by reviewers.

*Reviewer #1 (Recommendations for the authors):*

I found the manuscript well-written and easy to read. Nevertheless, I have some questions:

– Why did the authors use an agent-based model? Stochastic models (individual-based models) are mostly used for studying outbreaks and not in cases of disseminated diseases such as HPV. Do the authors consider that using a deterministic model could have provided different results?

– The authors mention in the Appendix that they considered an "all-or-nothing working mechanism for vaccine protection" for those HPV types included in the vaccine as well as for cross-protection. This should be added to the main article as it would be important for decision-makers who might not read the Appendix.

– Why did the authors consider that men did not develop natural immunity?

– Results, Pag 11 lines 152-153 "Among the four highlighted scenarios, the GO strategy with 60% coverage was also the least resilient". It would be better to start with the most resilient scenario.

– Presenting the prevented cases in the disrupted cohort (resilience) as a percentage of the cases prevented in the undisrupted scenarios would be more informative for the reader when comparing the four scenarios.

– The resilience ratios were constructed with the absolute number of cases prevented; they should have been constructed with the relative number of cases prevented because the values changed greatly between scenarios for the undisrupted cohorts. For example, the resilience ratio for GO 60% to GN 60% in the base case analysis for all states reported by the authors was 2.8 when using absolute numbers but it would be 2.4 when using relative numbers.

– Figure B2: The description says "with (A) high and (B) low cervical cancer incidence" but the figures are (A) all case (B) high incidence and (C) low incidence. Please correct.

*Reviewer #2 (Recommendations for the authors):*

1. Please describe the current status of HPV vaccination in India in the introduction, including when foreign HPV vaccines were available on the market in India, whether a routine HPV vaccination program has been / will be implemented in India (and when if yes), the current (estimated) HPV vaccine uptake, although I may miss the corresponding text.

2. Used an agent-based model (EpiMetHeos) to simulate the dynamic transmission of HPV infection of high-risk HPV types and a cervical cancer progression model (Altas) to estimate the impact on cervical cancer burden. The authors are suggested to include the advantage (and limitations) or the difference of using the agent-based model when compared with other models.

3. Model validity.

a. Please show the uncertainties (e.g., confidence interval) of the model calibration/fitting.

b. Please explicitly state the table(s)/figure(s) that contain(s) the values and assumptions used in the model/study in the manuscript, especially when referring to table(s)/figure(s) in the supplementary materials.

c. Percentile intervals (10% to 90%) were presented based on the simulations using the 100 parameter sets best fitting the sexual behaviour and HPV prevalence data obtained through calibration. Please state the criteria that were used to determine the "best fitting" (e.g., minimal difference, log-likelihood, or other metrics)

4. Vaccine characteristics

a. The study considered a locally developed quadrivalent HPV vaccine by referring the vaccine efficacy to Gardasil quadrivalent. The authors are advised to present the numerical values of the efficacy of the locally developed HPV vaccine explicitly in addition to claiming the similarity (although a reference was cited).

b. A sensitivity analysis on the below items may be useful when addressing the uncertainties of the assumption.

(i). shorter vaccine-induced protection, in addition to assuming no waning of protection over time.

(ii). vaccine efficacy, perhaps at the lower bound of the reported estimated efficacy, in addition to the point estimates.

c. Please discuss the latest estimate of vaccine uptake in India among girls and preferably among boys. That would be the comparator when evaluating the impact of a vaccination program. Please also provide the rationale/information for the four "highlighted" scenarios on vaccination coverage (lines 113-115).

d. Please state the year that the vaccination program would start in the model simulation.

5. Screening uptake in India

"Given the extremely low coverage of the existing screening programme in India, we have not considered the impact of screening in our study (Bruni et al., 2022)" (lines 142-143)

Please state the screening coverage explicitly.

6. Results

For each scenario, the authors are advised to include the relative changes in the number of cervical cancer cases eliminated with and without disruption.

7. There are likely some potential typos and grammar issues.

*Reviewer #3 (Recommendations for the authors):*

Overall

1. This study will benefit from a better selection of realistic vaccine trade-offs in moving from the base strategy (i.e GO 60%) to alternate strategies. This 'trade-off' is briefly mentioned in Lines 166-168: "Considering the required number of vaccine doses, we found a trade-off between dose-efficiency and resilience. In general, GO vaccination favoured dose efficiency and GN vaccination favoured resilience (Figure 1)." However, the trade-off is not easily interpreted when the alternate strategies use a different quantity of vaccines (as seen in Figure 1). If the alternate strategies used a similar quantity of vaccines, then the results of the study would be better understood by decision-makers/readers regarding the potential implications of various vaccination strategies with limited vaccine supplies.

2. The study will benefit from a lengthier discussion on the mechanisms through which gender-neutral vaccination can increase the HPV vaccine programme 'resilience' in preventing cervical cancer:

a) The mechanisms through which a gain in 'resilience' is achieved in moving to a GN strategy have been indirectly described in previous publications. During a period of vaccine disruption, my interpretation of the reason cervical cancer incidence increases is due to the reduction in vaccine coverage amongst all girls over time – and it has been shown in previous modelling studies that a gender-neutral vaccination programme is more beneficial when coverage amongst girls is lower. Therefore are the findings of this study a manifestation of this mechanism? – this would be worth discussing.

b) The authors describe the primary factor influencing resilience in Lines 241-243: "The main determinant underlying the gain in resilience by switching from GO to GN vaccination is the age difference between sexual partners, with men being on average older than women within sexual partnerships almost everywhere worldwide (Wellings et al., 2006)." However, it is not clear from the results of this paper why this is described as the 'main determinant'. Possibly providing evidence through sensitivity analyses to show how changes in age difference between sexual partners change the resilience of GN strategies may help prove the point to readers that this is the main determinant.

Methodology

3. The outcome of 'resilience' is calculated as the [in lines 128-131]: "mean number of cases still prevented across the birth cohorts with disruption, as a result of previous HPV vaccination,…, which was compared to the mean number of cases prevented across the birth cohorts before the disruption." I had a few concerns about this outcome measure:

a) My initial impression of the use of the term 'resilience' would be the ability of the vaccine programme to retain its population protective effect in times of disruption. Therefore the use of the 'ratio of the mean number of cases prevented' did not reflect resilience to me. Instead, I felt the description of the definition of 'resilience' in your Elfstrom et al. 2016 paper [referenced in lines 66-67] is clearer to readers as a measure of resilience: "the difference in percentage attributable to vaccination estimated with and without temporary coverage reduction". Would the team be able to reconcile whether the definitions/formulas used in this study and Elfstrom et al.'s 2016 paper are identical?

b) The calculation of the 'resilience ratio' in Appendix B1 is not immediately clear to readers, and an example/formula would help readers understand how it is calculated better.

c) The comparison of resilience ratios (seen in Appendix B1) would be better interpreted if the shift from GO 60% to alternative scenarios all involved the same quantity of vaccine supply. For e.g. resilience ratio in a (i) GO 60% to GO 90% scenario being compared with a (ii) GO 60% to Girl-60%/Male-30% scenario [assuming males & females are equal in each cohort, this would mean the same quantity of vaccines]. Such a comparison would be easier to interpret and would help decision-makers understand the benefits & drawbacks of either course of action.

Results

4. In lines 211-212 it is mentioned that: "coverage between 50% and 70% in girls might also be sufficient for elimination when combined with moderate coverage (30%) in boys." A table/graph to illustrate this scenario and its associated results would help readers understand the conclusion made here. If the data is presented in the appendix, then a reference to the location of the information in the appendix would be helpful.

Discussion

5. In lines 141-142 it is mentioned that: "we have not considered the impact of screening in our study". It might be useful to discuss the implications of this assumption in the discussion, bringing in the foreseeable changes in screening uptake & screening technology in India and other low-income countries.

6. The discussion would benefit from challenging the assumptions made in the model. Such as the assumptions that:

a) The total population in the model is kept constant over time

b) Only modelling heterosexual partnerships – this would not account for the fact that vaccinating 10-year-old boys who become predominantly men-who-have-sex-with-men (MSM) when sexually active, would likely have minimal to no impact on reducing cervical cancer incidence.

c) Sexual behaviours remain unchanged throughout the person's lifespan.

d) Susceptible-Infected-Susceptible (SIS) model used in men, which does have a period of immunity incorporated.

e) Having limited data on sexual behaviours and cervical cancer incidence in other Indian states.

---

## [Author Response]

Essential revisions:– Provide more context on HPV vaccination programs and screening in India.

India has not yet introduced HPV vaccination into the national immunization programme. Two Indian states have introduced state-wide HPV immunization programme under a girls-only strategy with high coverage (Sankaranarayanan et al., Lancet Oncol, 2019). Following the recent marketing authorisation granted to an indigenous vaccine (The Lancet, Editorial “HPV vaccination in south Asia: new progress, old challenges”, 2022), the prospect of introducing HPV vaccination into the national immunisation programme (NIP) has significantly improved. However, it is still unclear when the national introduction will take place and which HPV vaccine will be used. As for cervical cancer screening, the coverage of the existing screening programme in India is low, with an ever-in-life coverage of lower than 3% for women aged 30-49-years (Bruni et al., 2022).

We acknowledge that the manuscript would benefit from a more detailed description of the current status of HPV vaccination and cervical cancer screening in India. We have revised the manuscript in various places, as follow:

Page 4 lines 74-79: “HPV vaccination (in girls) has been introduced in two

Indian states with high vaccination coverage (Sankaranarayanan et al., 2019). Following the recent marketing authorisation granted to an indigenous vaccine (The Lancet, 2022), the prospect of introducing HPV vaccination into the national immunisation programme (NIP) has also significantly improved.”

Page 21 lines 389-392: “Recently, marketing authorisation has been awarded to a locally produced quadrivalent HPV vaccine in India … (The Lancet, 2022). … there is still uncertainty in which HPV vaccine will be used in the Indian immunisation programme, …”Page 4 lines 72-74: “Cervical cancer screening is still limitedly accessible in

India, with less than 3% ever-in-life coverage in women aged 30-49 years

(Sankaranarayanan et al. 2019, Bruni et al., 2022)”.

– Reviewers questioned the calculation methods used for resilience ratios. Consider providing more rationale for these ratios and/or using other metrics for measuring resilience.

In general, we think that resilience is a complex multifaceted phenomenon. Using different metrics may help revealing different aspects of the phenomenon.

In our study, the aspect we wanted to focus on was the absolute burden, i.e., how many cervical cancer cases we can prevent more during disruption when switching from one strategy/scenario to another. In our opinion, the absolute metric (the number of cervical cancer cases still prevented) is a straightforward metric for studying this aspect. Furthermore, considering this absolute metric would allow the evaluation of strategies in terms of whether it provides a minimum level of protection during disruption period.

To clarify why we have chosen the absolute metric, we have revised the manuscript as follows (page 22 lines 420-424): “As a measure of resilience, we used the mean number of cases still prevented, as a result of previous HPV vaccination, across the birth cohorts with disruption. Considering such a metric of resilience allows us to evaluate whether any given strategy provides a sufficient level of protection retained during disruption as compared to a possible prefixed target level of protection.”

Nevertheless, we agree with the reviewers that a relative metric of resilience, e.g., the proportion of cervical cancer cases still prevented as compared to the number of cases prevented before/without the disruption, could also be interesting. It provides an idea of how efficient the resource invested before disruption can be retained at disruption. However, in our opinion, it might be less straightforward to compare the “relative resilience” between strategy/scenario; an increase in relative resilience does not necessarily mean more cancer cases prevented during disruption.

Furthermore, we are unsure whether we should consider efficiency when dealing crises.

At the end, to avoid using too many different metrics, we chose to consider only the absolute resilience and the ratio thereof. Furthermore, the relative resilience can be easily read off from Figure 1 and calculated from Table 1.

Reviewer #1 (Recommendations for the authors):I found the manuscript well-written and easy to read. Nevertheless, I have some questions:– Why did the authors use an agent-based model? Stochastic models (individual-based models) are mostly used for studying outbreaks and not in cases of disseminated diseases such as HPV. Do the authors consider that using a deterministic model could have provided different results?

A similar question has also been posed by Reviewer #2. To our knowledge, the use of agent-based models is not limited to studying outbreaks. In fact, many calibrated models for studying HPV are also agent-based, e.g., HPVADVISE (http://www.marc-brisson.net/HPVadvise-LMIC.pdf) and HavardHPV (Burger et al., 2018, Vaccine). In general, agent-based models have the advantage of allowing long-term and overlapping partnerships in a more straightforward manner than in deterministic models. This aspect of modelling long-term and overlapping partnerships is particularly important for the Indian setting, where a considerable part of HPV transmission to married women is believed to occur through their marital partners. To clarify this, we have revised the manuscript as follows (page 20 lines 367368): “EpiMetHeos was used to simulate a dynamic sexual contact network, which allows long-term and overlapping partnerships, through which HPV infections were transmitted.”

– The authors mention in the Appendix that they considered an "all-or-nothing working mechanism for vaccine protection" for those HPV types included in the vaccine as well as for cross-protection. This should be added to the main article as it would be important for decision-makers who might not read the Appendix.

We have added the mention of the “all-or-nothing working mechanism for vaccine protection” in the main text, as follows (page 21 lines 402-404): “Vaccine protection was modelled through an all-or-nothing working mechanism, i.e., if the efficacy is X%, then X% of the vaccinated individuals are fully protected and 100-X% fully not protected.”

– Why did the authors consider that men did not develop natural immunity?

In men, it has generally been observed that the HPV prevalence and the rate of acquiring new HPV infection remain constant across age, and that the seroconversion rate is low after natural infection (Giuliano et al., Cancer Epidemiol Biomarkers Prev, 2008; Giuliano et al., Papillomavirus Res, 2015; Schiffman et al., Nat Rev Dis Primers, 2016). Hence, we assume no development of natural immunity in men.

We have revised the following sentence in the first paragraph of Appendix 1.1.3. to clarify this point: “Transmission of all HR HPV types are assumed to follow the “Susceptible-Infected-Removed/Immune-Susceptible” dynamics in women (Appendix 1-figure 1). In men, it has generally been observed that the HPV prevalence and the rate of acquiring new HPV infection remain constant across age, and that the seroconversion rate is low after natural infection (Giuliano et al., Cancer Epidemiol Biomarkers Prev, 2008; Giuliano et al., Papillomavirus Res, 2015; Schiffman et al., Nat Rev Dis Primers, 2016). Hence, we assumed the “Susceptible-Infected-Susceptible” dynamics in men.”

– Results, Pag 11 lines 152-153 "Among the four highlighted scenarios, the GO strategy with 60% coverage was also the least resilient". It would be better to start with the most resilient scenario.

We understand the reasoning of the reviewer’s suggestion. However, our main aim here is to illustrate and emphasize the gain in resilience by switching to GN strategy, therefore we prefer starting with the GO strategy.

– Presenting the prevented cases in the disrupted cohort (resilience) as a percentage of the cases prevented in the undisrupted scenarios would be more informative for the reader when comparing the four scenarios.

A similar point has been also raised by the Editor, please see our reply for the second point under Essential revision.

– The resilience ratios were constructed with the absolute number of cases prevented; they should have been constructed with the relative number of cases prevented because the values changed greatly between scenarios for the undisrupted cohorts. For example, the resilience ratio for GO 60% to GN 60% in the base case analysis for all states reported by the authors was 2.8 when using absolute numbers but it would be 2.4 when using relative numbers.

A similar point has been also raised by the Editor, please see our reply for the second point under Essential revision.

– Figure B2: The description says "with (A) high and (B) low cervical cancer incidence" but the figures are (A) all case (B) high incidence and (C) low incidence. Please correct.

We thank the reviewer for noticing the wrong labels. We have revised the text accordingly.

Reviewer #2 (Recommendations for the authors):1. Please describe the current status of HPV vaccination in India in the introduction, including when foreign HPV vaccines were available on the market in India, whether a routine HPV vaccination program has been / will be implemented in India (and when if yes), the current (estimated) HPV vaccine uptake, although I may miss the corresponding text.

A similar point has been also raised by the Editor, please see our reply for the first point under Essential revision.

2. Used an agent-based model (EpiMetHeos) to simulate the dynamic transmission of HPV infection of high-risk HPV types and a cervical cancer progression model (Altas) to estimate the impact on cervical cancer burden. The authors are suggested to include the advantage (and limitations) or the difference of using the agent-based model when compared with other models.

A similar question has also been posed by Reviewer #1. To our knowledge, the use of agent-based models is not limited to studying outbreaks. In fact, many calibrated models for studying HPV are also agent-based, e.g., HPVADVISE (http://www.marc-brisson.net/HPVadvise-LMIC.pdf) and HavardHPV (Burger et al., 2018, Vaccine). In general, agent-based models have the advantage of allowing long-term and overlapping partnerships in a more straightforward manner than in deterministic models. This aspect of modelling long-term and overlapping partnerships is particularly important for the Indian setting, where a considerable part of HPV transmission to married women is believed to occur through their marital partners. To clarify this, we have revised the manuscript as follows (page 20 lines 367368): “EpiMetHeos was used to simulate a dynamic sexual contact network, which allows long-term and overlapping partnerships, through which HPV infections were transmitted.”

3. Model validity.a. Please show the uncertainties (e.g., confidence interval) of the model calibration/fitting.

The uncertainties of the model calibration/fitting are shown in Appendix 1-figure 2 and Appendix 1-figure 3.

b. Please explicitly state the table(s)/figure(s) that contain(s) the values and assumptions used in the model/study in the manuscript, especially when referring to table(s)/figure(s) in the supplementary materials.

As stated in the Materials and methods section (page 20 lines 383-385), details on model calibration can be found in Appendices 1.1-3. To clarify which information can be found in which appendix section, we added the following specification: “See Appendix 1.1 for details on the structure of the HPV transmission model, 1.2 on the model calibration process and model fit, and 1.3 on the computation of the model outcomes.”

c. Percentile intervals (10% to 90%) were presented based on the simulations using the 100 parameter sets best fitting the sexual behaviour and HPV prevalence data obtained through calibration. Please state the criteria that were used to determine the "best fitting" (e.g., minimal difference, log-likelihood, or other metrics)

Calibration for determining the “best-fitting” parameters was based on log-likelihood as stated in the last paragraph of Appendix 1.2.3.

4. Vaccine characteristicsa. The study considered a locally developed quadrivalent HPV vaccine by referring the vaccine efficacy to Gardasil quadrivalent. The authors are advised to present the numerical values of the efficacy of the locally developed HPV vaccine explicitly in addition to claiming the similarity (although a reference was cited).

As stated in the Materials and methods section (page 21 line 389-391), the marketing authorisation was awarded to the locally produced vaccine based on successful immune-bridging between the new vaccine and the existing quadrivalent vaccine (Gardasil, MSD). The manuscript reporting these results is currently under review. Regarding vaccine efficacy, there are no data available yet.

b. A sensitivity analysis on the below items may be useful when addressing the uncertainties of the assumption.(i) Shorter vaccine-induced protection, in addition to assuming no waning of protection over time.(ii) Vaccine efficacy, perhaps at the lower bound of the reported estimated efficacy, in addition to the point estimates.

In this study, the aim is to explore the gain in resilience when switching from girls-only to gender-neutral strategy. Therefore, we focussed on sensitivity analyses regarding the variation in how disruption may occur. Furthermore, we have explored in depth the uncertainty in the duration of vaccine protection and lower efficacy in a previous publication, of which this study is the next (Man et al., Lancet Oncol, 2022). To clarify this, we have added the following to the revised manuscript (page 20 lines 371-373): “The models were previously calibrated to sexual behaviour, HPV prevalence, cervical cancer incidence in India to assess the impact of single-dose vaccination in India, while accounting for the uncertainty in the duration of vaccine protection (Man, Georges, de Carvalho, et al., 2022).”

c. Please discuss the latest estimate of vaccine uptake in India among girls and preferably among boys. That would be the comparator when evaluating the impact of a vaccination program. Please also provide the rationale/information for the four "highlighted" scenarios on vaccination coverage (lines 113-115).

As explained in the reply to the reviewer’s comment 1., India has not yet introduced a national immunization programme.

d. Please state the year that the vaccination program would start in the model simulation.

As it is not yet clear when India would start a national vaccination programme, we think it is more suitable to report the results in “years since the start of vaccination” instead of arbitrarily choosing a year of introduction.

5. Screening uptake in India"Given the extremely low coverage of the existing screening programme in India, we have not considered the impact of screening in our study (Bruni et al., 2022)" (lines 142-143)Please state the screening coverage explicitly.

We have revised the text to state the screening coverage explicitly, as follows (Page 4 lines 72-74): “Cervical cancer screening is still limitedly accessible in India, with less than 3% ever-in-life coverage in women aged 30-49 years (Sankaranarayanan et al. 2019, Bruni et al., 2022)”

6. ResultsFor each scenario, the authors are advised to include the relative changes in the number of cervical cancer cases eliminated with and without disruption.

A similar point has been also raised by the Editor, please see above our reply for the second point under Essential Revision.

7. There are likely some potential typos and grammar issues.

We have done our best to correct for typos and grammar issues throughout the manuscript.

Reviewer #3 (Recommendations for the authors):Overall1. This study will benefit from a better selection of realistic vaccine trade-offs in moving from the base strategy (i.e GO 60%) to alternate strategies. This 'trade-off' is briefly mentioned in Lines 166-168: "Considering the required number of vaccine doses, we found a trade-off between dose-efficiency and resilience. In general, GO vaccination favoured dose efficiency and GN vaccination favoured resilience (Figure 1)." However, the trade-off is not easily interpreted when the alternate strategies use a different quantity of vaccines (as seen in Figure 1). If the alternate strategies used a similar quantity of vaccines, then the results of the study would be better understood by decision-makers/readers regarding the potential implications of various vaccination strategies with limited vaccine supplies.

We agree with the reviewer that the efficiency of HPV vaccine dose allocation is a key operational issue and understand that the suggested strategies allow comparison of scenarios in terms of how to best allocate resources. However, we still decided to focus on comparing strategies of girls-only and gender-neutral that do not necessarily correspond to the same amount of vaccine doses. The reason was that while the policymakers have control to decide on the vaccination strategy (either girls-only or gender-neutral), they usually do not have control over the coverage eventually attained in the population. For example, the scenario of girls-only vaccination with 90% coverage requires the same number of vaccines as the scenario of gender-neutral vaccination with 60% in girls and 30% in boys. However, it would not be realistic to stop vaccinating at 30% coverage in boys if more boys would like to be vaccinated. While data on uptake in boys is still scarce, there are indication that uptake in boys could be expected to be similar to that in girls (Wähner et al., 2023, Infection). This was the rationale behind the chosen scenarios. In our opinion, it is enough to demonstrate the trade-off between dose-efficiency and resilience with the part of the Results section mentioned by the reviewer.

2. The study will benefit from a lengthier discussion on the mechanisms through which gender-neutral vaccination can increase the HPV vaccine programme 'resilience' in preventing cervical cancer:a) The mechanisms through which a gain in 'resilience' is achieved in moving to a GN strategy have been indirectly described in previous publications. During a period of vaccine disruption, my interpretation of the reason cervical cancer incidence increases is due to the reduction in vaccine coverage amongst all girls over time – and it has been shown in previous modelling studies that a gender-neutral vaccination programme is more beneficial when coverage amongst girls is lower. Therefore are the findings of this study a manifestation of this mechanism? – this would be worth discussing.

Resilience in cervical cancer prevention is the herd effect in women unvaccinated due to disruption because the reduced population force of infection from men that have contact with these unvaccinated women. We agree with the reviewer that vaccinating more girls could also decrease the force of infection in men, and so increasing resilience. However, vaccinating boys is a more direct way to reduce the force of infection in men than vaccinating girls. This is likely the mechanism underlying the higher resilience achieved by gender-neutral strategy than girls-only strategy.

To better explain this mechanism, we have added the following sentences to the Discussion section (page 15 lines 275-278): “As showed by our results, increasing coverage in girls also helps to increase resilience. However, vaccinating men is a more direct way to reduce the force of infection in men than vaccinating girls, which is likely the mechanism underlying the higher resilience achieved by gender-neutral strategy than girls-only strategy.”

b) The authors describe the primary factor influencing resilience in Lines 241-243: "The main determinant underlying the gain in resilience by switching from GO to GN vaccination is the age difference between sexual partners, with men being on average older than women within sexual partnerships almost everywhere worldwide (Wellings et al., 2006)." However, it is not clear from the results of this paper why this is described as the 'main determinant'. Possibly providing evidence through sensitivity analyses to show how changes in age difference between sexual partners change the resilience of GN strategies may help prove the point to readers that this is the main determinant.

We acknowledge that the wording “main determinant” was not appropriate. We have rephrased it as follows: “An important determinant underlying the gain in resilience by switching from GO to GN vaccination is likely the age difference between sexual partners.”

Methodology3. The outcome of 'resilience' is calculated as the [in lines 128-131]: "mean number of cases still prevented across the birth cohorts with disruption, as a result of previous HPV vaccination,.., which was compared to the mean number of cases prevented across the birth cohorts before the disruption." I had a few concerns about this outcome measure:a) My initial impression of the use of the term 'resilience' would be the ability of the vaccine programme to retain its population protective effect in times of disruption. Therefore the use of the 'ratio of the mean number of cases prevented' did not reflect resilience to me. Instead, I felt the description of the definition of 'resilience' in your Elfstrom et al. 2016 paper [referenced in lines 66-67] is clearer to readers as a measure of resilience: "the difference in percentage attributable to vaccination estimated with and without temporary coverage reduction". Would the team be able to reconcile whether the definitions/formulas used in this study and Elfstrom et al.'s 2016 paper are identical?

We think that the definition in this study is in line with the initial impression the reviewer describes as the definition in Elfström et al. What the reviewer refers to as “protective effect retained” is what we refer to as “cervical cancer cases still prevented”, which is what we define as “resilience” in this study.

The ratio of the number of cervical cases still prevented, or resilience ratio, is then used to evaluate the increase in resilience between scenarios. We have now clarified the definition of the resilience both in the Materials and methods section and in the caption of Table 1 and Appendix 2-Table 1. – Page 22 lines 424-425: “Estimates of resilience were compared between the highlighted scenarios A-D based on their ratio.”

Page 36 lines 637-639: “[Resilience ratio of scenario X to scenario Y] is defined is [resilience of scenario Y] / [resilience of scenario X]. For example, in the base case, [resilience ratio of GO 60% to GO 90%] = [resilience of GO 90%] / [resilience of GO 60%] = 209 / 107 = 2.0.”

Finally, what we define as resilience in Elfström et al. corresponds to the “drop in cervical cancer prevented” in Figure 1. In this study, we defined resilience to focus on the protective effect still retained during disruption.

b) The calculation of the 'resilience ratio' in Appendix B1 is not immediately clear to readers, and an example/formula would help readers understand how it is calculated better.

The following definition of the resilience ratio is now added to the caption of Table 1 and Appendix 2-Table 1: “[Resilience ratio of scenario X to scenario Y] is defined is [resilience of scenario Y] / [resilience of scenario X].

For example, in the base case, [resilience ratio of GO 60% to GO 90%] = [resilience of GO 90%] / [resilience of GO 60%] = 209 / 107 = 2.0.”.

c) The comparison of resilience ratios (seen in Appendix B1) would be better interpreted if the shift from GO 60% to alternative scenarios all involved the same quantity of vaccine supply. For e.g. resilience ratio in a (i) GO 60% to GO 90% scenario being compared with a (ii) GO 60% to Girl-60%/Male-30% scenario [assuming males & females are equal in each cohort, this would mean the same quantity of vaccines]. Such a comparison would be easier to interpret and would help decision-makers understand the benefits & drawbacks of either course of action.

See above for our reply to Reviewer 3, comment 1.

Results4. In lines 211-212 it is mentioned that: "coverage between 50% and 70% in girls might also be sufficient for elimination when combined with moderate coverage (30%) in boys." A table/graph to illustrate this scenario and its associated results would help readers understand the conclusion made here. If the data is presented in the appendix, then a reference to the location of the information in the appendix would be helpful.

Figure 4 is the figure illustrating this scenario and the associated results. This figure has been referred to in the line right before the lines the reviewer has mentioned. To help the reader understand we are still considering the results in Figure 4, we have added the reference to this figure at the end of these lines, as follows: “For instance, coverage between 50% and 70% in girls might also be sufficient for elimination when combined with moderate coverage (30%) in boys (Figure 4).”

Discussion5. In lines 141-142 it is mentioned that: "we have not considered the impact of screening in our study". It might be useful to discuss the implications of this assumption in the discussion, bringing in the foreseeable changes in screening uptake & screening technology in India and other low-income countries.

To our opinion, it is still difficult to forecast how much coverage of cervical cancer screening would increase in India. Therefore, we assumed no change in screening uptake in our analysis. To account for this point, we added the following in the Discussion (page 17 lines 319-320): “We also did not consider changes in cervical cancer screening, as it is difficult to predict how much coverage of cervical cancer screening will increase in India in the coming years.”

6. The discussion would benefit from challenging the assumptions made in the model. Such as the assumptions that:a) The total population in the model is kept constant over time

Keeping the total population constant is a common practice in this type of HPV transmission model, which is essential to ensure stable model impact estimates, such as reduction in HPV prevalence and cervical cancer risk.

b) Only modelling heterosexual partnerships – this would not account for the fact that vaccinating 10-year-old boys who become predominantly men-who-have-sex-with-men (MSM) when sexually active, would likely have minimal to no impact on reducing cervical cancer incidence.

While it is true that our model does not consider homosexual partnerships among MSM, we do not think the impact on cervical cancer incidence would be much affected by this assumption. This is because of the likely small proportion of MSM in India (estimated as 3% in the 2006 NACO report of sexual behaviour), and vaccination in 10-year-old boys is expected to result in similar in boys who later become heterosexual or MSM. However, we would be fitting to discuss the direct benefit in MSM for HPV related cancer in men. We have now added this point to the Discussion section, as follows (page 18 lines 349-352): “In some context, important arguments for choosing GN strategy could be … the direct health benefits in men who have sex with men.”

c) Sexual behaviours remain unchanged throughout the person's lifespan.

We did allow sexual behaviours to change throughout the person’s lifespan. This was explained in the second paragraph of Appendix 1.1.2. (“Formation probabilities of stable as well as one-off partnerships depend on the individual’s sex, age group and the assigned risk group of sexual activity.”) and showed in Appendix 1-figure 2.

d) Susceptible-Infected-Susceptible (SIS) model used in men, which does have a period of immunity incorporated.

In men, it has generally been observed that the HPV prevalence and the rate of acquiring new HPV infection remain constant across age, and that the seroconversion rate is low after natural infection (Giuliano et al., Cancer Epidemiol Biomarkers Prev, 2008; Giuliano et al., Papillomavirus Res, 2015; Schiffman et al., Nat Rev Dis Primers, 2016). Hence, we assume no development of natural immunity in men.

We have revised the following sentence in the first paragraph of Appendix 1.1.3. to clarify this point: “Transmission of all HR HPV types are assumed to follow the “Susceptible-Infected-Removed/Immune-Susceptible” dynamics in women (Appendix 1-figure 1). In men, it has generally been observed that the HPV prevalence and the rate of acquiring new HPV infection remain constant across age, and that the seroconversion rate is low after natural infection (Giuliano et al., Cancer Epidemiol Biomarkers Prev, 2008; Giuliano et al., Papillomavirus Res, 2015; Schiffman et al., Nat Rev Dis Primers, 2016). Hence, we assumed the “Susceptible-Infected-Susceptible” dynamics in men.”

e) Having limited data on sexual behaviours and cervical cancer incidence in other Indian states.

The availability and limitations of data on sexual behaviours and cervical cancer incidence across Indian states were discussed in detail in our previous publication in which we presented the calibrated model for the first time (Man et al., Lancet Oncol, 2022). To account for this, we have added the following sentence to the Discussion (page 16 lines 294-296): “The availability and limitations of data on sexual behaviours and cervical cancer incidence across Indian states were discussed in detail in our previous publication in which we presented the calibrated model for the first time (Man et al., Lancet Oncol, 2022).”